# HYPERPARAMETERS IN CONTINUAL LEARNING: A REALITY CHECK WITH CLASS-INCREMENTAL LEARNING

## ABSTRACT

Continual learning (CL) aims to train a model on a sequence of tasks (*i.e.*, a CL scenario) while balancing the trade-off between plasticity (learning new tasks) and stability (retaining prior knowledge). The dominantly adopted conventional evaluation protocol for CL algorithms selects the best hyperparameters within a given scenario and then evaluates the algorithms using these hyperparameters in the same scenario. However, this protocol has significant shortcomings: it overestimates the CL capacity of algorithms and relies on unrealistic hyperparameter tuning, which is not feasible for real-world applications. From the fundamental principles of evaluation in machine learning, we argue that the evaluation of CL algorithms should focus on assessing the generalizability of their CL capacity to unseen scenarios. Based on this, we propose the Generalizable Two-phase Evaluation Protocol consisting of a hyperparameter tuning phase and an evaluation phase. Both phases share the same scenario configuration (*e.g.*, number of tasks) but are generated from different datasets. Hyperparameters of CL algorithms are tuned in the first phase and applied in the second phase to evaluate the algorithms. We apply this protocol to class-incremental learning, both with and without pretrained models. Across more than 8,000 experiments, our results show that most state-of-the-art algorithms fail to replicate their reported performance, highlighting that their CL capacity has been significantly overestimated in the conventional evaluation protocol.

## 1 INTRODUCTION

In recent years, extensive research has been conducted on continual learning (CL) with the goal of effectively learning knowledge from a sequence of tasks (Wang et al., 2023). A neural network model in such CL scenarios faces a crucial trade-off between learning new knowledge from novel tasks (plasticity) and maintaining knowledge on previous tasks (stability) (Mermillod et al., 2013). To address this inherent trade-off, numerous algorithms have been proposed for successful CL in various domains (Wang et al., 2023). In these domains, many CL studies have focused on classification, primarily concentrating on class-incremental learning (class-IL) (Masana et al., 2020) without or with pretrained models (Zhou et al., 2024a). However, deploying CL algorithms requires careful hyperparameter tuning. Figure 1 illustrates the conventional evaluation protocol (including hyperparameter tuning) dominantly employed in both offline and online class-incremental learning (Zhou et al., 2022; Boschini et al., 2022; Zhou et al., 2024b; Smith et al., 2023; Seo et al., 2024). Additionally, similar evaluation protocols are also widely applied across other CL domains for semantic segmentation (Cha et al., 2021b; Yuan & Zhao, 2024), test-time adaptation (Yoo et al., 2024; Lee et al., 2024), federated learning (Piao et al., 2024), self-supervised learning (Fini et al., 2022; Cha et al., 2024) and large language models (Ke et al., 2023; Wu et al., 2024).

Many algorithms have been considered state-of-the-art based on performance validated through the conventional evaluation protocol. However, this raises two issues: First, the hyperparameter tuning method used in this protocol is not applicable to real-world CL scenarios. Second, it results in evaluation overfitting to a given scenario and dataset, which in turn leads to an overestimation of their CL capacity. In other words, this protocol only assesses performance in a seen scenario but fails to evaluate generalizability to new, unseen ones—an essential aspect for real-world applications. While several alternative evaluation protocols and hyperparameter tuning methods have been proposed, they

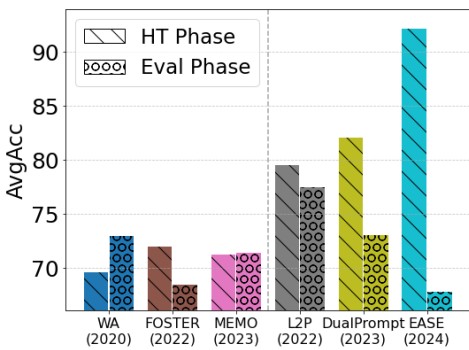

Figure 1: This figure illustrates the conventional evaluation protocol. First, a CL scenario is constructed using a benchmark dataset, where each task has its own training, validation, and test sets. Second, to find the best hyperparameters $\mathcal{H}^*$, a model is sequentially trained up to the final task using the sampled hyperparameters. After training for each task $t$, the model $\theta_t$ is evaluated using the validation dataset. This process is repeated for various hyperparameters, and the best hyperparameters are selected based on performance. Finally, the model is trained using the CL algorithm with the best hyperparameters $\mathcal{H}^*$ in the same CL scenario, and report the evaluation result on the test dataset. Note that in many studies, the results are reported using only $D_{\text{val}}$ without separating $D_{\text{te}}$ (*i.e.*, $D_{\text{te}} = D_{\text{val}}$).

also have limitations: 1) they require to tune additional hyperparameters for their methods (Delange et al., 2021; Liu et al., 2023), or 2) they are only applied to a few old algorithms, and have not gained widespread acceptance (Chaudhry et al., 2018b; Chen et al., 2023; Bornschein et al., 2023). As a result, the issues with the conventional evaluation protocol have been largely ignored, and it remains the dominant evaluation protocol for evaluating CL algorithms until now.

In this paper, we aim to reveal the limitation of the conventional evaluation protocol by revisiting the fundamental principles of evaluation in machine learning. From this perspective, we argue that the evaluation of CL algorithms should prioritize assessing the generalizability of each algorithm's CL capacity across unseen scenarios.

To achieve this goal, we propose a revised evaluation protocol, the Generalizable Two-phase Evaluation Protocol (GTEP), which consists of two phases: the hyperparameter tuning phase and the evaluation phase. Both phases share the same CL scenario configuration (*e.g.*, the number of tasks and classes per task) yet leverage distinct datasets. In the hyperparameter tuning phase, a model is trained using various hyperparameters of an algorithm, and the best hyperparameters are selected based on performance. These best hyperparameters are then applied directly to train the model using the algorithm in the evaluation phase, where the measured performance serves as a reliable benchmark for the algorithm's CL capacity in unseen

Figure 2: Results on both phases.

scenarios. As an initial application of this protocol, we focus on the most actively researched domain of CL—class-incremental learning (class-IL)—both with and without pretrained models (Wang et al., 2023). From approximately 8,000 experiments, we derive the following key findings:

- First, as shown in Figure 2, most state-of-the-art class-IL algorithms achieve superior performance in the hyperparameter tuning (HT) phase, which is almost identical the conventional evaluation protocol. However, they reveal limited generalizability in their CL capacity in the evaluation (Eval) phase. This tendency is particularly pronounced in the recently proposed algorithms.
- Second, further analysis shows that these algorithms are limited by long training times, a large number of required parameters, or significant performance variance, suggesting they are less efficient than expected.

Based on extensive experimental results with the proposed evaluation protocol, we highlight major shortcomings of the conventional approach, which consistently overestimates the CL capacity of algorithms. In conclusion, we advocate for a fundamental reconsideration of the evaluation protocol across all domains to drive meaningful progress in CL research.

## 2 RELATED WORK

**Continual learning**   Continual learning (CL) research has been conducted in various domains (Wang et al., 2023; Parisi et al., 2019; Delange et al., 2021; Masana et al., 2020). In the beginning, the CL research focus on task-incremental learning (Parisi et al., 2019; Delange et al., 2021), exploring diverse approaches (Li & Hoiem, 2017; Aljundi et al., 2018; Chaudhry et al., 2018a; Cha et al., 2021a; Yoon et al., 2017). As the field progressed, attention shifted to the more challenging scenario, class-incremental learning (class-IL) (Masana et al., 2020). This shift leads to the investigation of exemplar-based methods, involving the effective utilization of exemplar memory storing a subset of the dataset from previous tasks (Rebuffi et al., 2017; Zhao et al., 2020; Cha et al., 2023a). Since then, using the exemplar memory has become standard, with several methods building on this foundation. Regularization-based methods, which overcome catastrophic forgetting by introducing a novel regularization (Wu et al., 2019; Douillard et al., 2020), and model expansion-based methods, which dynamically expand model capacity to balance the trade-off between stability and plasticity, have become the most powerful approach, achieving state-of-the-art performance (Wang et al., 2022b; Yan et al., 2021; Zhou et al., 2022; Wang et al., 2022a).

Class-IL using pretrained models has recently gained considerable attention for achieving strong performance without relying on the exemplar memory (Zhou et al., 2024a). Prompt-based methods enable class-IL through prompt learning while keeping the pretrained model frozen. These approaches have evolved over time, incorporating techniques such as using prompt pool (Wang et al., 2022d), prompt combination (Wang et al., 2022c), decomposed prompt (Smith et al., 2023), and prompt generation (Jung et al., 2023). Additionally, representation-based methods derive class prototypes from the pretrained model and use them for classification (Zhou et al., 2023b). To enhance the separability of these prototypes, several recent methods have focused on reducing class-wise correlation (McDonnell et al., 2024; Zhou et al., 2024b).

**Evaluation and hyperparameter tuning of CL**   Several papers have proposed new evaluation metrics and protocols for the proper assessment of CL algorithms in classification. Traditionally, accuracy-based metrics (e.g., final and average accuracy) have been used as the primary metrics of evaluating performance of CL algorithms (Parisi et al., 2019; Masana et al., 2020; Chaudhry et al., 2018a). However, recent studies have highlighted limitations of these metrics, particularly regarding computational costs (Prabhu et al., 2023) and learned representations (Cha et al., 2023b). Delange et al. (2021) introduced a hyperparameter tuning method for task-incremental learning, which involves first conducting a maximum plasticity search and then selecting the best hyperparameters using stability decay. Similarly, Liu et al. (2023) proposed a hyperparameter selection method for class-IL based on a bandit algorithm. However, both approaches entail additional training costs and the need to tune extra hyperparameters. Other studies have proposed evaluation protocols similar to ours (Chaudhry et al., 2018b; Chen et al., 2023; Bornschein et al., 2023). However, these protocols have only been applied to a limited number of older algorithms in specific domains, which fails to fully uncover the limitations of the conventional evaluation protocol. In addition to these efforts, despite discussions on proper CL evaluation (Mundt et al., 2022), the conventional evaluation protocol has continued to dominate the assessment of state-of-the-art CL algorithms across various domains.

## 3 TOWARDS EVALUATING THE GENERALIZABILITY OF THE CL CAPACITY

### 3.1 MOTIVATION: IMPROPER HYPERPARAMETER TUNING

As shown in Figure 1, the primary flaw of the conventional evaluation protocol is that it optimizes an algorithm's hyperparameters in a given CL scenario and then evaluates the algorithm using those same hyperparameters. Surprisingly, many studies have reported their results by directly tuning hyperparameters on test data without considering separate validation sets (*i.e.*, set $D_{te}^{HT} = D_{val}^{HT}$), as seen in studies such as Wu et al. (2019); Douillard et al. (2020); Zhao et al. (2020); Yan et al. (2021); Wang et al. (2022b); Zhou et al. (2022); Wang et al. (2022a;d); Zhou et al. (2023b; 2024b), and others. Note that this approach is only feasible in experimental scenarios where all task data is always available. Consequently, this hyperparameter tuning method fails to capture the real challenges of CL and is not applicable to real-world situations. While many studies partially address this limitation by reporting robust performance across various experiments with some fixed or minimally adjusted hyperparameters (Wang et al., 2022a;d; Zhou et al., 2024b), these evaluations are still based on given

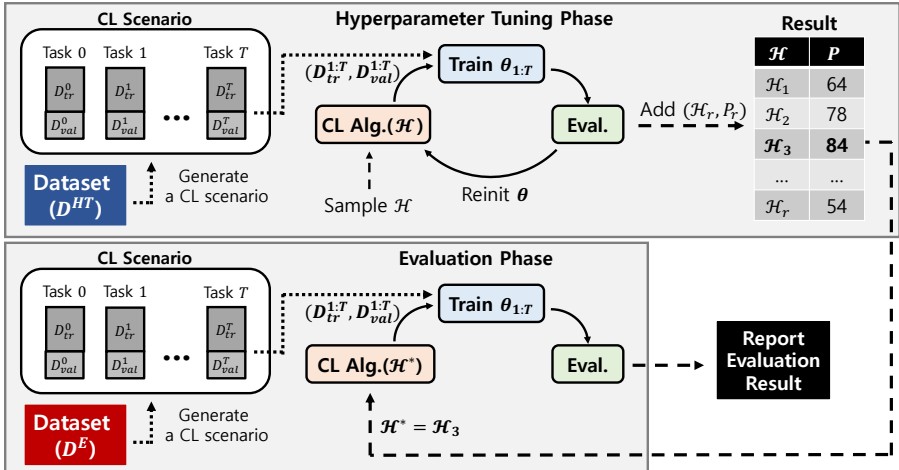

Figure 3: Illustration of the proposed evaluation protocol. Both phases share the same CL scenario configuration (*e.g.*, the number of tasks and number of classes in each task) but they are generated from distinct datasets ($D^{HT}$ and $D^E$). Best hyperparameters are selected in the hyperparameter tuning phase. Then, the evaluation phase access a CL algorithm by training a model using them. Note that evaluating an algorithm solely based on the results from the hyperparameter tuning phase is identical to the conventional evaluation protocol without using $D_{te}$.

scenarios (*i.e.*, seen scenarios), making it challenging to assess whether the algorithms would perform equally well in unseen scenarios. Nevertheless, this conventional protocol remains the predominant evaluation protocol for assessing algorithms across most CL domains.

---

**Algorithm 1:** The Generalizable Two-phase Evaluation Protocol

---

**Input** : A CL algorithm $\mathcal{A}$, a model $\theta$, the dataset for the hyperparameter tuning phase $D^{HT}$, the dataset for the evaluation phase $D^E$, the number of random samplings $R$, the number of trials $S$, and the number of hyperparameters $K$.
**Output** : Final evaluation result ($P^E$) for a CL algorithm $\mathcal{A}$ in the evaluation phase

    1. $\{(\mathcal{H}_i, P_i^{HT})\}_{i=1}^R \leftarrow \texttt{HyperparameterTuning}(\theta, \mathcal{A}, D^{HT}, R, S, K)$

    2. $\mathcal{H}^* \leftarrow \texttt{SelectBestHyperparameter}(\{(\mathcal{H}_i, P_i^{HT})\}_{i=1}^R)$

    3. $P^E \leftarrow \texttt{Evaluation}(\theta, \mathcal{A}, D^E, \mathcal{H}^*, S)$

---

### 3.2 GENERALIZABLE TWO-PHASE EVALUATION PROTOCOL (GTEP) FOR CL

Given the previously discussed issues with the conventional evaluation protocol, the key question becomes: What hyperparameter tuning and evaluation protocol should be used to properly assess CL algorithms? Note that effective evaluation in machine learning should prioritize realistic methods tailored to each learning scenario, rather than rigidly adhering to assumptions (*e.g.*, i.i.d.) for theoretical convenience. In this regards, we argue that evaluating the generalizability of each algorithm's CL capacity is essential. For example, consider a real-world CL scenario where an algorithm is applied to a CL scenario consisting of a sequence of tasks. Since the entire task data would not be fully accessible at once, the conventional hyperparameter tuning method cannot be applied. In such cases, a reasonable approach is to construct a simulated CL scenario, reflecting the expected actual CL scenario, using a benchmark or available dataset. This involves identifying the best hyperparameters in the simulated scenario and then applying them to the actual CL scenario. In other words, one of the basic evaluation protocols—consistent with the fundamental principles of evaluation in machine learning—is to tune hyperparameters in seen scenarios (*e.g.*, simulated scenarios) and test them in unseen scenarios (*e.g.*, actual scenarios).

Building on the above concept, we propose a revised evaluation protocol consisting of two phases, the Generalizable Two-phase Evaluation Protocol (GTEP): hyperparameter tuning and evaluation. Figure 3 and Algorithm 1 outlines the overall process. The key idea is that CL scenarios for the hyperparameter tuning and evaluation phases are generated from different datasets (*i.e.*, $D^{HT} \neq D^E$) but share the same scenario configuration (*e.g.*, the number of tasks and classes per task), based

on expectations on the actual scenario. In the hyperparameter tuning phase, the goal is to identify the best hyperparameters for the CL algorithm. In the evaluation phase, these hyperparameters are applied to assess the algorithm's CL capacity in unseen scenarios, providing a more realistic measure of its generalizability.

The pseudo algorithm of the hyperparameter tuning phase is outlined in Algorithm 2 of the Appendix. First, we randomly sample hyperparameters $h_k$ from a predefined set $h_k^{Set}$ and build a list of selected hyperparameters $\mathcal{H}_r$. Next, we generate a predefined CL scenario using the function $\mathcal{F}$ with shuffled class orderings. Afterward, the model $\theta$ is trained using the selected hyperparameters $\mathcal{H}_r$, the CL algorithm $\mathcal{A}$, and the training dataset $D_{tr}^{HT}$. Performance ($P^{HT}$) is then measured on the validation dataset $D_{val}^{HT}$. This phase returns multiple sets of hyperparameters and their corresponding performance. Next, using the SelectBestHyperparameter function in Algorithm 1, we select the best hyperparameters, denoted as $\mathcal{H}^*$. Note that the hyperparameter tuning phase is identical to the conventional evaluation protocol. However, we only use the results from this phase to select the best hyperparameters.

In the evaluation phase (shown in Algorithm 3 of the Appendix), we train a model $\theta$ using the CL algorithm with the best hyperparameters $\mathcal{H}^*$. The trained model is then tested on the validation dataset $D_{val}^E$. The final performance metric is the averaged performance ($P^E$) of the trained model across multiple class orderings, which serves as the evaluation criterion for the CL algorithm.

To find the best hyperparameters for each algorithm, we optimize both algorithm-specific hyperparameters (*e.g.*, regularization strength) and general hyperparameters (*e.g.*, learning rate and batch size). During the hyperparameter tuning phase, we train the model with $R$ sets of randomly selected hyperparameters and account for $S$ task orderings per set. In the evaluation phase, we assess the performance across $S$ task orderings as well. In this paper, we set $R = 30$ and $S = 5$ for all experiments. We also take into account various similarity scenarios between the hyperparameter tuning dataset ($D^{HT}$) and the evaluation dataset ($D^E$). High similarity indicates that the characteristics of the dataset used in the actual scenario are somewhat predictable, allowing us to generate a scenario for the hyperparameter tuning phase using a similar dataset. Conversely, low similarity suggests unpredictability, indicating that the datasets used to generate scenarios in both phases differ significantly. Evaluating each algorithm under both similarity cases offers a comprehensive understanding of the generalizability of its CL capacity. Furthermore, these efforts towards accurate evaluation highlight the methodological differences from previously proposed evaluation protocols (Chaudhry et al., 2018b; Chen et al., 2023; Bornschein et al., 2023; Mundt et al., 2022), as the revised evaluation protocol. Additionally, note that the high-level concept of the proposed protocol can be applied to various CL domains by considering the specific characteristics of these domains (*e.g.*, imbalanced classes per task, blurred task boundaries, or entirely different domains such as semantic segmentation) in the CL scenario generation process (denoted as $\mathcal{F}$ in Algorithms 2 and 3) for both phases.

## 4 EXPERIMENTAL RESULTS

In this section, we present extensive experimental results using our proposed protocol in the most actively researched domain of continual learning (CL) (Wang et al., 2023), class-incremental learning (class-IL) both without and with pretrained models (Masana et al., 2020; Zhou et al., 2024a; 2023a).

### 4.1 CLASS-INCREMENTAL LEARNING WITHOUT PRETRAINED MODELS

**Experimental settings** We conduct the hyperparameter tuning and evaluation phases using benchmark datasets, as shown in Table 1. From ImageNet-1k (Deng et al., 2009), we derive two subsets, ImageNet-100-1 and ImageNet-100-2, each containing 100 randomly selected non-overlapping classes. To account for varying dataset similarities, we further divide CIFAR-100 (Krizhevsky et al., 2009) and ImageNet-100-1 into disjoint classes, generating CIFAR-50-1, CIFAR-50-2, ImageNet-50-1, and ImageNet-50-2. We focus on two primary class-incremental learning (class-IL) scenarios (Masana et al., 2020): **10 Tasks**, where the model learns an equal number of classes from each task, and **6**

Table 1: Scenarios and datasets.

| Scenario | $D^{HT}$ | $D^E$ |
|---|---|---|
| 10 Tasks (C10×T10) | ImageNet-100-1 | ImageNet-100-2 |
| 6 Tasks (C50×T1 + C10×T5) | | |
| 10 Tasks (C5×T10) | ImageNet-50-1, CIFAR-50-1 | ImageNet-50-2, CIFAR-50-2 |
| 6 Tasks (C25×T1 + C5×T5) | | |

**Tasks**, where the model learns half of the total classes in the first task then evenly distributes the remaining classes evenly across subsequent tasks. Note that evaluating using both scenarios has been widely considered the proper assessment of each algorithm (Masana et al., 2020; Zhou et al., 2023a) The table presents the configuration of the number classes (C) and tasks (T) for each scenario. We conduct experiments using ResNet (He et al., 2016). We employ two key performance metrics commonly used for evaluating class-IL algorithms (Masana et al., 2020): **Acc** is final classification accuracy for the entire validation dataset after training the final task, and **AvgAcc** $= \frac{1}{T} \sum_{t=1}^{T} Acc_t$, where $Acc_t$ denotes accuracy on the validation data up to task $t$. The hyperparameters that yield the highest **harmonic mean** of Acc and AvgAcc are selected during the hyperparameter tuning phase.

**Baselines** We evaluate nine major class-IL algorithms, including replay-based methods (Replay, iCaRL (Rebuffi et al., 2017), and WA (Zhao et al., 2020)) and regularization-based methods (BiC (Wu et al., 2019) and PODNet (Douillard et al., 2020)) and expansion-based methods (DER (Yan et al., 2021), FOSTER (Wang et al., 2022b), and BEEF (Wang et al., 2022a)). Note that we use the partially implemented DER, as neither PyCIL nor the official DER code includes the implementation details for masking and pruning. Replay serves as a naive baseline, where a model is fine-tuned using both the exemplar memory and the current task's dataset. Note that these algorithms have demonstrated progressively improved performance in the order of their publication. Among them, FOSTER, BEEF, and MEMO are recognized as the current *state-of-the-art*, reporting superior performance that surpasses DER by a small margin. We conduct experiments using the implementation code proposed in PyCIL (Zhou et al., 2023a). The size of the exemplar memory is set to 2000 for ImageNet-100, and 1000 for ImageNet-50 and CIFAR-50 variants. More details on settings, predefined hyperparameter sets and selected hyperparameters are presented in Section B.1 of the Appendix.

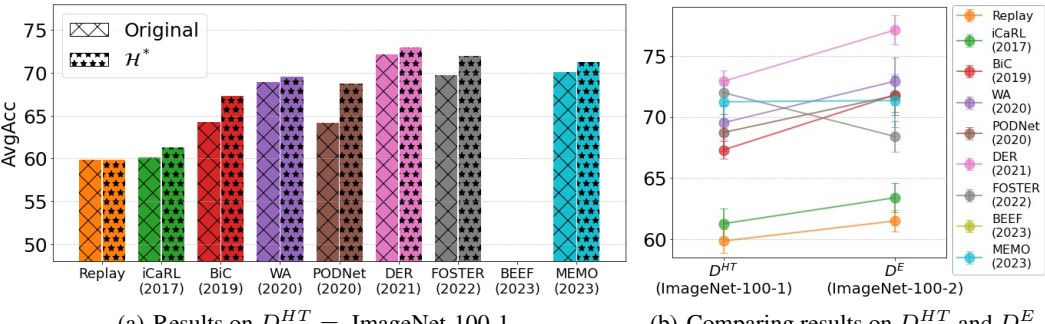

(a) Results on $D^{HT}$ = ImageNet-100-1      (b) Comparing results on $D^{HT}$ and $D^E$

Figure 4: Experimental results (AvgAcc) on the 10 Tasks scenario using ImageNet-100-1 for $D^{HT}$ and ImageNet-100-2 for $D^E$ (high similarity). The term 'Original' and $\mathcal{H}^*$ refer to the use of reported hyperparameters and hyperparameters selected from our protocol, respectively. BEEF constantly returns NaN in training loss at specific seeds so we could not report its performance.

**Experiments using original and selected hyperparameters** To demonstrate whether the hyperparameters identified during the hyperparameter tuning phase achieve better performance than those previously reported, we conduct experiments with both sets of hyperparameters. Figure 4(a) presents results on $D^{HT}$ = ImageNet-100-1, showing that using the best hyperparameters ($\mathcal{H}^*$) generally outperforms the original ones across all algorithms except BEEF. Note that the performance differences among DER, FOSTER, and MEMO are within their respective standard deviations. Considering the hyperparameter tuning phase aligns with the conventional evaluation protocol, this graph indicates that each algorithm reflects the performance trends observed in their respective papers, gradually improving over time in accordance with the order of publication. On the other hand, we confirm that BEEF is significantly sensitive to hyperparameters, as it occasionally results in NaN (Not a Number) in training loss for specific seeds, even when using the original hyperparameters.

In the evaluation phase, we apply the best hyperparameters to assess the CL capacity in unseen scenarios generated by $D^E$. Note that, due to differences in the datasets between these phases, the final performance may vary across phases, even when using identical hyperparameters for each algorithm. Figure 4(b) presents experimental results. The graph shows that the CL capacity of the state-of-the-art algorithms (*i.e.*, FOSTER, BEEF, and MEMO) is significantly inferior to that of older algorithms, such as WA, BiC and PODNet. Additionally, BEEF again produces NaN values for certain seeds. In contrast, DER demonstrates superior generalizability of its CL capacity, consistently maintaining strong performance in both phases.

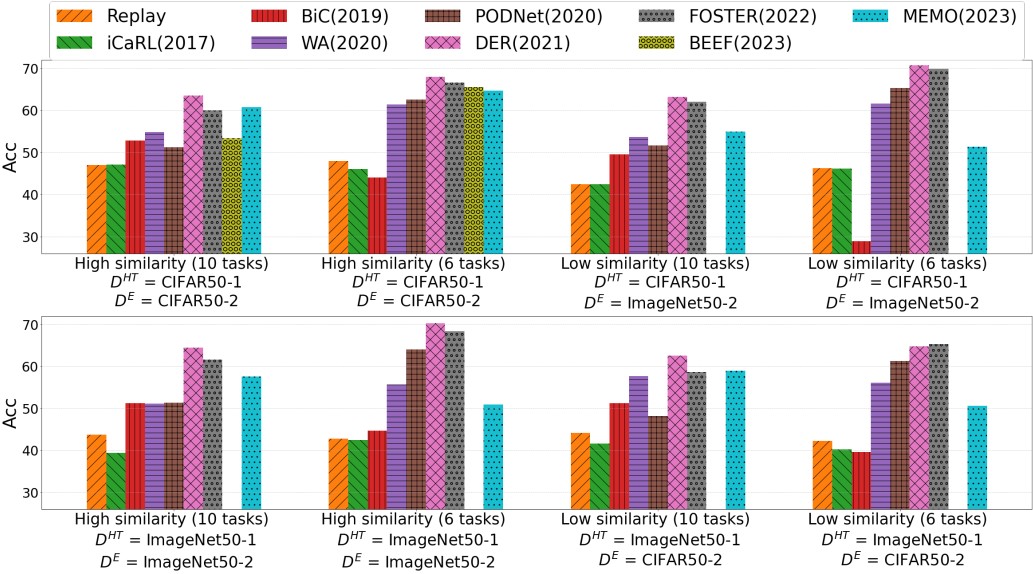

Figure 5: Bar graphs depict the experimental results from the evaluation phase. The Y-axis represents final classification accuracy (Acc). The parentheses next to each algorithm indicate the publication year. The bar graphs in the first row show the experimental results using the best hyperparameters selected in the hyperparameter tuning phase with $D^{HT}$ = CIFAR-50-1 , while the graphs in the second row show the results using $D^{HT}$ = ImageNet-50-1 . In cases of using ImageNet-50-1 or ImageNet-50-2, we encountered challenges in obtaining results for BEEF due to NaN issues.

**Experiments across diverse similarity cases** To broadly assess the generalizability of each algorithm's CL capacity, we conduct experiments across various similarity cases. The bar graphs in the first row of Figure 5 display results for both high and low similarity cases, using the best hyperparameters selected during the hyperparameter tuning phase using $D^{HT}$ = CIFAR-50-1. In most cases, iCaRL performs worse than Replay, and BiC also struggles in some cases (*e.g.*, 6 tasks in both high and low similarity settings). Additionally, both WA and PODNet outperform other regularization-based methods, with PODNet particularly excelling in the 6 Tasks. Lastly, the current state-of-the-art methods—FOSTER, BEEF, and MEMO—exhibit lower performance compared to DER, with BEEF again showing significant sensitivity, especially on ImageNet-50-2.

The second row of Figure 5 presents results using the best hyperparameters selected based on $D^{HT}$ = ImageNet-50-1. The trends are consistent with previous experiments: DER maintains superior performance in most cases, although FOSTER surpasses DER in the low similarity case (6 tasks). Additionally, BEEF suffers from NaN issues in training loss for certain seeds.

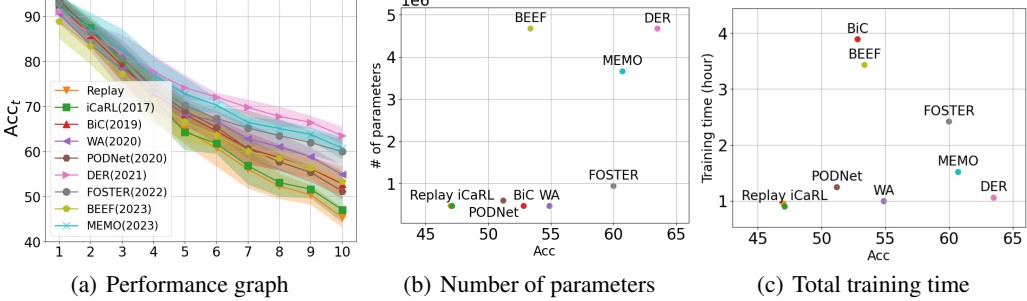

(a) Performance graph      (b) Number of parameters      (c) Total training time

Figure 6: Experimental analysis in the evaluation phase. All experimental results are obtained by first identifying the best hyperparameters using CIFAR-50-1 (10 Tasks) in the hyperparameter tuning phase, then evaluating each algorithm using CIFAR-50-2 (10 Tasks) in the evaluation phase. (b) and (c) show results after training up to the final task.

**Additional analysis** Figure 6(a) shows the evaluation results for each task $t$ in the evaluation phase, with shaded areas representing the standard deviation across 5 trials. From these graphs, it is evident that DER consistently outperforms current state-of-the-art algorithms (*i.e.*, FOSTER, BEEF and MEMO). Considering the standard deviation, the performances of FOSTER and MEMO are nearly indistinguishable. Among the remaining algorithms, WA demonstrates relatively better performance while BEEF performs similarly to the order algorithms.

Recent studies have increasingly focused on evaluating CL algorithms based on their training costs, particularly in terms of GPU usage and energy consumption (Prabhu et al., 2023; Chavan et al., 2023). However, these evaluations were often conducted by either limiting the number of training iterations or comparing costs under a fixed number of training epochs. Building on this, we extend the analysis by examining the final model size and total training time for each algorithm, using their best hyperparameters to ensure a fair and comprehensive comparison of efficiency. Figures 6(b) and 6(c) present scatter plots showing achieved accuracy, total parameter counts, and training times. DER performs the best and requires relatively less training time. Nevertheless, it exhibits considerable inefficiency in the total number of parameters, which increases linearly with the number of tasks, raising concerns about its actual cost-efficiency as a CL algorithm. On the other hand, BiC, BEEF, and MEMO fail to demonstrate superior performance while requiring similar or longer training times compared to DER, highlighting their serious inefficiency.

## 4.2 CLASS-INCREMENTAL LEARNING WITH PRETRAINED MODELS

**Experimental details** We conduct both the hyperparameter tuning and evaluation phases using widely used datasets in class-incremental learning (class-IL) with pretrained models, including CUB-200 (Wah et al., 2011), ImageNet-R (Hendrycks et al., 2021a), and ImageNet-A (Hendrycks et al., 2021b), all of which contain 200 classes. To explore diverse similarity cases, we divide these datasets into two disjoint subsets, as outlined in Table 2. Following Sun et al. (2023), we consider two major class-IL scenarios: **20 Tasks** and **10 Tasks**, where the model learns an equal number of

Table 2: Scenarios and datasets.

| Scenario | $D^{HT}$ | $D^E$ |
|---|---|---|
| 20 Tasks (C10×T20) | CUB-200, ImageNet-R | ImageNet-R, ImageNet-A |
| 10 Tasks (C20×T10) | | |
| 20 Tasks (C5×T20) | CUB-100-1, ImageNet-R-1 | CUB-100-2, ImageNet-R-2, ImageNet-A-2 |
| 10 Tasks (C10×T10) | | |

classes in each task. Note that the 20 Tasks scenario has been commonly regarded as the standard for better evaluating algorithm performance due to the need to handle more tasks. For all experiments, we employ the ViT B16 model, which is pretrained on ImageNet (Dosovitskiy et al., 2020). The best hyperparameters are selected based on the same metrics: the **harmonic mean** of **Acc** and **AvgAcc**.

**Baselines** We select six major algorithms: prompt-based methods (L2P (Wang et al., 2022d), DualPrompt (Wang et al., 2022c) and CODA-Prompt (Smith et al., 2023)) and representation-based methods (Adam-Adapter (Zhou et al., 2023b), Ranpac (McDonnell et al., 2024) and EASE (Zhou et al., 2024b)). Within each category, CODA-Prompt and EASE represent current *state-of-the-art* algorithms. Although DAP (Jung et al., 2023) reports better performance within the prompt-based method category, we exclude it due to fairness issues in comparison, as mentioned in Zhou et al. (2024a). All experiments are conducted using code implemented in PILOT (Sun et al., 2023). Details on experimental settings, predefined hyperparameter sets the best hyperparameters are proposed in Section B.2 of the Appendix.

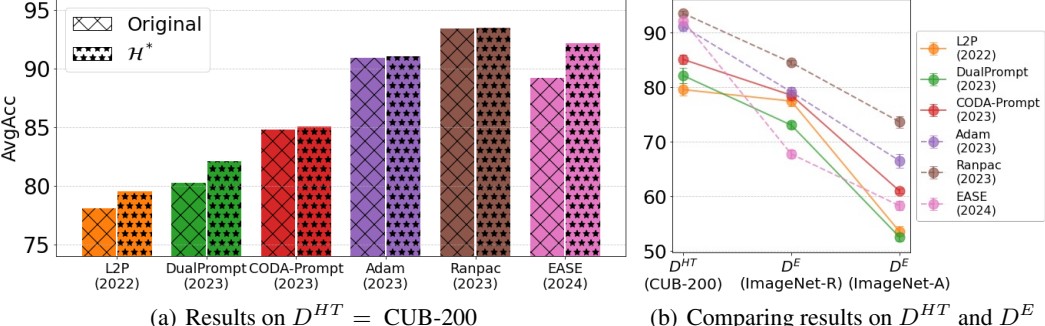

(a) Results on $D^{HT}=$ CUB-200      (b) Comparing results on $D^{HT}$ and $D^E$

Figure 7: Experimental results (AvgAcc) for 10 Tasks scenario using CUB-200 for $D^{HT}$, ImageNet-R, and ImageNet-A for $D^E$ (low similarity). The term 'Original' and $\mathcal{H}^*$ refer to the use of original hyperparameters and the hyperparameters selected from our protocol, respectively.

**Experiments using original and selected hyperparameters** To verify best hyperparameters selected in the hyperparameter tuning phase, we conduct experiments on $D^{HT}=$ CUB-200 using both the reported and selected hyperparameters of each algorithm. Figure 7(a) demonstrates that using the selected hyperparameters leads to better performance across all algorithms. Additionally,

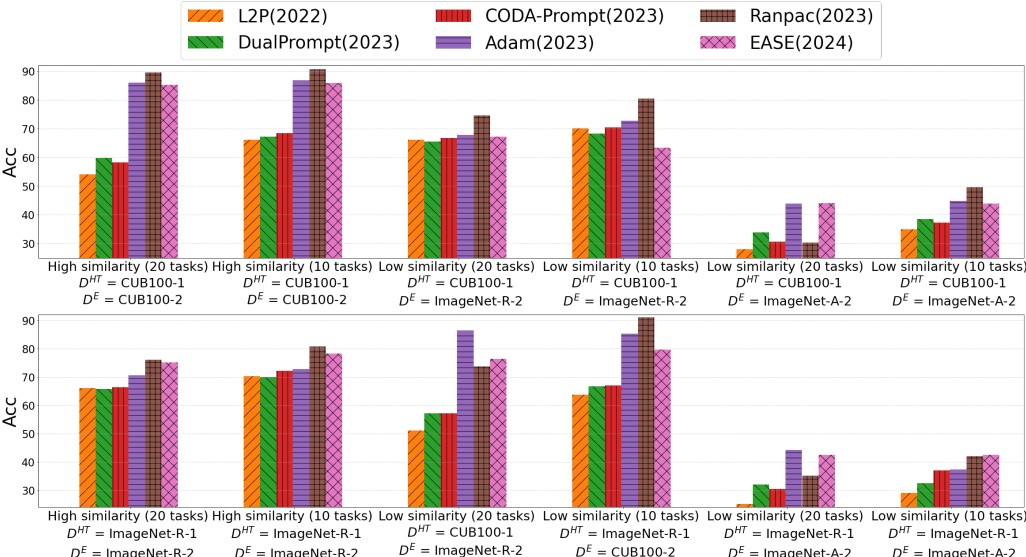

Figure 8: Bar graphs depict the experimental results from the evaluation phase. The Y-axis represents final accuracy (Acc). In the legend, the parentheses next to each algorithm indicate the publication year. The bar graphs in the first row show the experimental results using the best hyperparameters selected in the hyperparameter tuning phase with $D^{HT}$ = CUB100-1 , while the graphs in the second row display the results using $D^{HT}$ = ImageNet-R-1 .

we observe that the performance of each algorithm gradually improves in accordance with their publication order, as reported in the respective papers. Note that Ranpac and EASE demonstrate similar performance, with differences falling within their standard deviations.

Following our evaluation protocol, we apply the best hyperparameters for each algorithm in the evaluation phase. We conduct experiments for two evaluation phases using ImageNet-R and ImageNet-A as $D^E$ and Figure 7(b) shows experimental results. From these results, we can confirm the following observations: First, among the prompt-based algorithms (solid lines), DualPrompt exhibits degraded performance compared to L2P in both evaluation phases. Additionally, CODA-Prompt demonstrates superior performance in all cases, although it shows nearly identical performance to L2P in the ImageNet-R. In the case of the representation-based algorithms (dashed lines), Ranpac consistently maintains its superiority across all datasets. However, EASE, recognized as the current state-of-the-art, shows significantly poorer performance in both evaluation phases.

**Experiments across diverse similarity cases** Figure 8 presents the experimental results evaluated in the evaluation phase. Similar to trends reported in Zhou et al. (2024a), representation-based methods generally outperform prompt-based methods. However, significant differences are observed under the proposed evaluation protocol: First, the prompt-based methods have reported substantial performance improvements over previous algorithms (*e.g.*, 7-10% increases on the CUB200 dataset for each algorithm (Zhou et al., 2024a)). However, the proposed evaluation protocol reveals either no significant performance difference between them (*e.g.*, low similarity (20 tasks) using ImageNet-R-2 in the first row of the graph) or cases where an order algorithm outperforms a newer one (*e.g.*, high similarity (20 tasks) using CUB100-2 in the first row of the graph). Second, the current state-of-the-art representation-based method, EASE, often underperforms compared to Ranpac, especially in low similarity cases (*e.g.*, low similarity (10 tasks) using ImageNet-R-2 in the first row of the graph). Lastly, while Ranpac achieves the best performance in most cases, it exhibits significantly degraded performance in several low similarity cases (*e.g.*, low similarity (20 tasks) using ImageNet-A-2 in the first row of the graph). This degradation is attributed to considerable performance instability in certain tasks.

**Additional analysis** As we already confirmed in the previous experiments, Figure 9(a) illustrates that Ranpac suffers from significant instability in certain tasks, resulting in a substantial increase in standard deviation (shaded area). Furthermore, we observe that the state-of-the-art algorithms, EASE and CODA-Prompt in their respective categories, do not consistently outperform baseline algorithms like ADAM and DualPrompt in many cases, highlighting a lack of generalizability in their CL capacity.

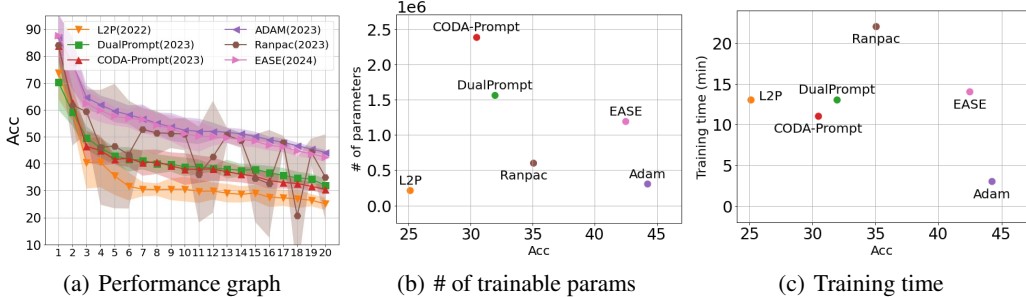

(a) Performance graph  (b) # of trainable params  (c) Training time

Figure 9: Experimental analyses in the evaluation phase. All experimental results are obtained by first identifying the best hyperparameters using ImageNet-R-1 (20 Tasks) in the hyperparameter tuning phase, then evaluating each algorithm using ImageNet-A-2 (20 Tasks) in the evaluation phase. (b) and (c) show the results after training up to the final task.

Figures 9(b) and 9(c) display the number of trainable parameters and training times with the best hyperparameters. For prompt-based algorithms, training times are comparable; however, CODA-Prompt requires more parameters while delivering lower performance compared to DualPrompt. Among representation-based methods, the oldest algorithm (*i.e.*, ADAM) achieves the best performance with minimal costs in terms of trainable parameters and training time.

In Section C of the Appendix, we present additional experimental results, including training graphs and numerical data related to the results discussed in the manuscript.

## 5 CONCLUDING REMARKS

**Problems with the conventional evaluation protocol**  The conventional evaluation protocol, which is predominantly used to assess continual learning (CL) algorithms, has significant flaws because it fails to consider real-world situations. In particular, the hyperparameter tuning method, which relies on repeated training in a given scenario is not only inapplicable to real-world CL scenarios but also tends to overestimate the CL capacity of each algorithm. According to the fundamental evaluation principles of machine learning, the evaluation of CL algorithms should prioritize assessing the generalizability of their CL capacity to unseen scenarios. In this regard, we propose the Generalizable Two-phase Evaluation Protocol (GTEP), which involves tuning hyperparameters in seen scenarios and then applying them to unseen scenarios, considering various similarity cases.

**Summary of experimental findings**  Our experiments across various similarity cases provide several key insights: First, the CL capacity of many algorithms, especially recent ones, has been significantly overestimated. Although most state-of-the-art algorithms perform well in seen scenarios, their CL capacity to unseen scenarios is often lacking. Second, we found that some of these algorithms are highly sensitive to hyperparameters, resulting in instances where they fail to learn specific task orders or exhibit significant performance variance on certain tasks. These two findings indicate that they have reported overfitted results to the seen scenarios under the conventional evaluation protocol, raising serious questions about their real-world applicability. Finally, even algorithms that perform relatively well in the proposed protocol often incur excessive costs (*e.g.*, training time and number of parameters), undermining one of the key objectives of continual learning: cost-efficiency. Although we reported the experimental results in class-incremental learning, we argue that these issues can naturally be inferred to occur in other CL domains that use the same conventional evaluation protocol.

**How should we evaluate going forward? – Key Takeaways**  We believe that the proposed evaluation protocol provides a fundamental approach to assess the generalizability of each algorithm's CL capacity, taking into account both the fundamental evaluation principles in machine learning and its real-world applications. Therefore, to make meaningful progress in CL research, we suggest that future evaluations across all CL domains should at least check the following:

- Does the proposed algorithm outperform baseline algorithms when the best hyperparameters selected from the hyperparameter tuning phase are applied to the evaluation phase?
- In the evaluation phase, is the proposed algorithm more efficient in terms of training costs (*e.g.*, total parameters and training time) compared to baseline algorithms? Additionally, does it avoid significant instability?

There are several limitations to our work and we discuss both limitations and future work in Section D of the Appendix.

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

# A    ALGORITHM TABLES

---

**Algorithm 2:** Pseudo algorithm of the hyperparameter tuning phase

---

**Input** : A CL algorithm $\mathcal{A}$, a model $\theta$, the dataset for the hyperparameter tuning phase $D^{HT}$, the number of random samplings $R$, the number of trials $S$, the number of hyperparameters $K$, and the function that generates a CL scenario $\mathcal{F}$.

**Output** : $\{(\mathcal{H}_i, P_i^{HT})\}_{i=1}^R$

    1. Result $\leftarrow \{\}$

    2. **for** $r \leftarrow 1\ to\ R$ **do**

       3.   **for** $k \leftarrow 1\ to\ K$ **do**

       4.     $h_k \leftarrow \text{RandomSample}(h_k^{\text{Set}})$

       5.   $\mathcal{H}_r \leftarrow (h_0, h_1, \cdots, h_K)$

       6.   **for** $s \leftarrow 1\ to\ S$ **do**

       7.     Initialize $\theta$

       8.     $D_{tr}^{HT}, D_{val}^{HT} \leftarrow \mathcal{F}(\text{Shuffle}(D^{HT}))$

       9.     $P_s^{HT} \leftarrow \text{TrainCL}(\mathcal{A}, D_{tr}^{HT}, D_{val}^{HT}, \theta, \mathcal{H}_r)$

    10.   $P_r^{HT} \leftarrow \frac{1}{S} \sum_{s=1}^S P_s^{HT}$

    11.   Add $(\mathcal{H}_r, P_r^{HT})$ to Result

---

---

**Algorithm 3:** Pseudo algorithm of the evaluation phase

---

**Input** : A CL Algorithm $\mathcal{A}$, a model $\theta$, the dataset for the Eval phase $D^E$, the best hyperparameter value $\mathcal{H}^*$, the number of trials $S$, the number of hyperparameters $K$, and the function that generates a CL scenario $\mathcal{F}$.

**Output** : Final evaluation result $(P^E)$ for $\mathcal{A}$

    1. **for** $s \leftarrow 1\ to\ S$ **do**

    2.   Initialize $\theta$

    3.   $D_{tr}^E, D_{val}^E \leftarrow \mathcal{F}(\text{Shuffle}(D^E))$

    4.   $P_s^E \leftarrow \text{TrainCL}(\mathcal{A}, D_{tr}^E, D_{val}^E, \theta, \mathcal{H}^*)$

    5. $P^E \leftarrow \frac{1}{S} \sum_{s=1}^S P_s^E$

---

# B ADDITIONAL DETAILS ON EXPERIMENTAL SETTINGS

## B.1 CLASS-INCREMENTAL LEARNING WITHOUT A PRETRAINED MODEL

**Experimental details** We conduct all experiments using PyCIL (Zhou et al., 2023a) in the following environment: Python 3.8, PyTorch 1.13.1, and CUDA 11.7. We use ResNet-18 and ResNet-32 architectures for our experiments. For class-incremental learning without a pretrained model, we employ the SGD optimizer with a momentum of 0.9 across all methods, consistent with their respective implementations. Other hyperparameters, however, are sampled during the hyperparameter tuning phase.

Table 3: Hyperparameters for training the first task.

| Hyperparameters | Values |
|---|---|
| Init epochs | 200 |
| Init learning rate | 0.1 |
| Init milestones | [60, 120, 170] (Only applied when 'StepLR' is selected) |
| Init learning rate decay | 0.1 |
| Init weight decay | 0.0005 |

**Predefiend hyperparameters** Recent studies have demonstrated that newer algorithms perform better when trained for more epochs on the first task and fewer epochs on subsequent tasks (Masana et al., 2020). Additionally, it is known that performance on the first task significantly impacts overall performance (Cha et al., 2023b). To apply this approach consistently across all algorithms, we train a model on the first task using the hyperparameters listed in Table 3. Subsequently, we train that model with randomly sampled hyperparameters starting from the second task.

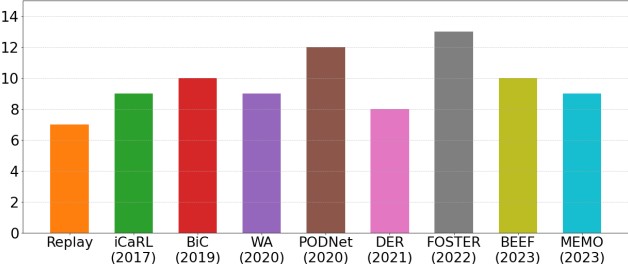

Figure 10: # of hyperparameters.

Figure 10 shows the number of hyperparameters for each algorithm. We consider both algorithm-specific and general hyperparameters in the hyperparameter tuning phase. Table 4 presents the sets of predefined hyperparameters considered for each algorithm. Note that 'Epoch', 'Num milestones', 'LR decay', 'Batch size', 'Weight decay', and 'LR scheduler' are commonly considered hyperparameters for all algorithms. Additionally, both 'Num milestones' and 'Lr decay' are applicable only when 'StepLR' is selected as a scheduler. The others are specific hyperparameters of each algorithm. We consider all the hyperparameters necessary for implementing each algorithm. For instance, even if a specific algorithm uses the same value for a particular hyperparameter across all experiments (*e.g.*, fixing the strength of an additional regularization to 1), we aimed to find the best hyperparameter for it (*e.g.*, setting the strength as $\alpha$ and finding the best value of it in the hyperparameter tuning phase). We determine the range of values for the predefined hyperparameters based on the following criteria. First, for general hyperparameters, we establish the range to include all optimal values reported by each algorithm. For specific hyperparameters related to each algorithm, we not only include the optimal values report in the papers but also considered the full range of values that were explored during their hyperparameter searches.

When the LR scheduler is set to StepLR, the milestones must be determined. To achieve this, we generalize the process of random sampling based on the milestones used in existing algorithms. First, we randomly sample num_milestones. Based on this sampling, the milestones for the StepLR are set according to the following rule: For example, if Num_milestones is set to 2, the milestones are defined as [epoch*(2/5), epoch*(4/5)]. If set to 3, the milestones become [epoch*(2/7), epoch*(4/7), epoch*(6/7)]. Similarly, for 4 milestones, the values are [epoch*(2/9), epoch*(4/9), epoch*(6/9),

epoch*(8/9)]. However, note that the num_milestones is ignored when another LR schduler is selected.

Table 4: The predefined set of hyperparametes for class-IL without a pretrained model.

| Algorithm | Hyperparameter Name | $h^{Set}$ |
|---|---|---|
| All algorithms | Epoch | [30, 70, 120, 160, 200] |
| | LR | [0.05, 0.1, 0.15, 0.2, 0.3] |
| | Num milestones | [2, 3, 4] |
| | LR decay | [0.1, 0.3, 0.5] |
| | Batch size | [32, 64, 128, 256, 512] |
| | Weigh decay | [0.0001, 0.0005, 0.001, 0.005] |
| | LR Scheduler | ['StepLR', 'Cosine'] |
| iCaRL, BiC, WA and FOSTER | T (KD) | [0.5, 1, 1.5, 2, 2.5] |
| BiC, WA and FOSTER | $\lambda$ (KD) | [0.5, 1, 1.5, 2, 3] |
| BiC | Split ratio | [0.05, 0.1, 0.15, 0.2, 0.3] |
| iCaRL, PODNet, DER and MEMO | $\lambda$ (Aux) | [0.5, 1, 1.5, 2, 3] |
| FOSTER | $\lambda$ (FE) | [0.5, 1, 1.5, 2, 3] |
| FOSTER | $\beta_1$ | [0.93, 0.95, 0.97, 0.99] |
| FOSTER | $\beta_2$ | [0.93, 0.95, 0.97, 0.99] |
| PODNet | Num proxy | [10, 20, 30, 50, 100] |
| PODNet, FOSTER and BEEF | Post FT epochs | [5, 10, 20, 30, 50] / [30, 70, 120, 160, 200] (FOSTER and BEEF) |
| PODNet | Post FT LR | [0.001, 0.003, 0.005, 0.007, 0.01] |
| PODNet | Adaptive factor | [True, False] |
| BEEF | Energy weight | [0.001, 0.005, 0.01, 0.02, 0.05] |
| BEEF | Logit alignment | [1.1, 1.4, 1.7, 2.0, 2.3] |
| MEMO | Exemplar batch size | [16, 32, 64, 128, 256] |

**Original hyperparameters** The following shows the original hyperparameters of each algorithm reported in PyCIL.

- Replay: ep_70_lr_0.1_lr_decay_0.1_batch_128_w_decay_0.0002_scheduler_steplr
- BiC: ep_170_lr_0.1_lr_decay_0.1_batch_128_w_decay_0.0002_scheduler_steplr T_2_lambda_kd_0_split_ratio_0.1
- PODNet: ep_160_milestone_2_lr_0.1_lr_decay_0.1_batch_128_w_decay_0.0005_scheduler_cosine lambda_c_5_lambda_f_1.0_nb_proxy_10_ft_epochs_20_ft_lrate_0.005_adaptive_factor_True
- FOSTER: ep_170_lr_0.1_lr_decay_0.1_batch_128_w_decay_0.0005_scheduler_cosine T_2_lambda_kd_1_fe_1_beta_0.96_0.97_comp_ep_130
- MEMO: ep_170_milestone_3_lr_0.1_lr_decay_0.1_batch_128_w_decay_0.0002_scheduler_steplr lambda_aux_1_examplar_bs_64
- iCaRL: ep_170_lr_0.1_lr_decay_0.1_batch_128_w_decay_0.0002_scheduler_steplr T_2_lambda_aux_1
- WA: ep_170_milestone_3_lr_0.1_lr_decay_0.1_batch_128_w_decay_0.0002_scheduler_steplr T_2.0_lambda_kd_0
- DER: ep_170_lr_0.1_lr_decay_0.1_batch_128_w_decay_0.0002_scheduler_steplr lambda_aux_1
- BEEF: ep_170_milestone_4_lr_0.1_lr_decay_0.1_batch_128_w_decay_0.0005_scheduler_cosine fusion_ep_60_energy_w_0.01_logits_align_1.7

Note that setting 'lambda_kd = 0' for both BiC and WA indicates the use of their adaptive rule.

**Best hyperparameters (ImageNet-100, 10 Tasks)** The following represents the best hyperparameters of each algorithm selected in the hyperparameter tuning phase using ImageNet-100 (10 Tasks).

- Replay: ep_70_milestone_3_lr_0.2_lr_decay_0.1_batch_64_w_decay_0.0001_scheduler_steplr
- BiC: ep_120_milestone_3_lr_0.1_lr_decay_0.1_batch_32_w_decay_0.0001_scheduler_steplr
  T_1_lambda_kd_3.0_split_ratio_0.1
- PODNet: ep_30_milestone_4_lr_0.05_lr_decay_0.1_batch_64_w_decay_0.0001_scheduler_steplr
  lambda_c_3_lambda_f_1.5_nb_proxy_20_ft_epochs_5_ft_lrate_0.005_adaptive_factor_False
- FOSTER: ep_30_milestone_4_lr_0.05_lr_decay_0.1_batch_64_w_decay_0.0001_scheduler_steplr
  T_1_lambda_kd_1.5_fe_1_beta_0.93_0.97_comp_ep_160
- MEMO: ep_120_milestone_3_lr_0.15_lr_decay_0.1_batch_512_w_decay_0.001_scheduler_steplr
  ambda_aux_0.5_examplar_bs_32
- iCaRL: ep_200_milestone_3_lr_0.15_lr_decay_0.1_batch_64_w_decay_0.0001_scheduler_cosine
  T_2.5_lambda_aux_2
- WA: ep_120_milestone_3_lr_0.1_lr_decay_0.1_batch_32_w_decay_0.0001_scheduler_steplr
  T_1_lambda_kd_3.0_split_ratio_0.1
- DER: ep_200_milestone_3_lr_0.15_lr_decay_0.1_batch_64_w_decay_0.0001_scheduler_cosine
  lambda_aux_3
- BEEF: ep_120_milestone_2_lr_0.2_lr_decay_0.3_batch_128_w_decay_0.0001_scheduler_steplr
  fusion_ep_30_energy_w_0.02_logits_align_2.3

**Best hyperparameters (ImageNet-100, 6 Tasks)** The following represents the best hyperparameters of each algorithm selected in the hyperparameter tuning phase using ImageNet-100 (6 Tasks).

- Replay: ep_70_milestone_3_lr_0.2_lr_decay_0.1_batch_64_w_decay_0.0001_scheduler_steplr
- BiC: ep_30_milestone_4_lr_0.05_lr_decay_0.1_batch_64_w_decay_0.0001_scheduler_steplr
  T_1_lambda_kd_1.5_split_ratio_0.1
- PODNet: ep_30_milestone_2_lr_0.15_lr_decay_0.1_batch_128_w_decay_0.001_scheduler_steplr
  lambda_c_9_lambda_f_0.5_nb_proxy_100_ft_epochs_10_ft_lrate_0.007_adaptive_factor_False
- FOSTER: ep_70_milestone_3_lr_0.05_lr_decay_0.1_batch_512_w_decay_0.0001_scheduler_cosine
  T_2.5_lambda_kd_0.5_fe_3_beta_0.95_0.93_comp_ep_30
- MEMO: ep_120_milestone_3_lr_0.15_lr_decay_0.1_batch_512_w_decay_0.001_scheduler_steplr
  lambda_aux_0.5_examplar_bs_32
- iCaRL: ep_200_milestone_3_lr_0.15_lr_decay_0.1_batch_64_w_decay_0.0001_scheduler_cosine
  T_2.5_lambda_aux_2
- WA: ep_170_lr_0.1_lr_decay_0.1_batch_128_w_decay_0.0002_scheduler_steplr
  T_2_lambda_kd_1
- DER: ep_200_milestone_3_lr_0.15_lr_decay_0.1_batch_64_w_decay_0.0001_scheduler_cosine
  lambda_aux_3
- BEEF: ep_30_milestone_4_lr_0.05_lr_decay_0.1_batch_128_w_decay_0.0001_scheduler_steplr
  fusion_ep_70_energy_w_0.01_logits_align_1.4

**Best hyperparameters (CIFAR-50, 10 Tasks)** The following represents the best hyperparameters of each algorithm selected in the hyperparameter tuning phase using CIFAR-50 (10 Tasks).

- Replay: ep_160_milestone_3_lr_0.15_lr_decay_0.3_batch_32_w_decay_0.0001_scheduler_cosine
- BiC: ep_200_milestone_2_lr_0.1_lr_decay_0.1_batch_32_w_decay_0.0001_scheduler_cosine
  T_0.5_lambda_kd_0.5_split_ratio_0.2
- PODNet: ep_70_milestone_2_lr_0.1_lr_decay_0.1_batch_32_w_decay_0.0001_scheduler_steplr
  lambda_c_1_lambda_f_1_nb_proxy_10_ft_epochs_30_ft_lrate_0.007_adaptive_factor_False
- FOSTER: ep_120_milestone_3_lr_0.1_lr_decay_0.5_batch_32_w_decay_0.0001_scheduler_steplr
  T_2_lambda_kd_1.5_fe_0.5_beta_0.97_0.93_comp_ep_160

- MEMO: ep_120_milestone_4_lr_0.05_lr_decay_0.3_batch_32_w_decay_0.0005_scheduler_steplr lambda_aux_0.5_examplar_bs_16

- iCaRL: ep_70_milestone_3_lr_0.05_lr_decay_0.3_batch_32_w_decay_0.001_scheduler_cosine T_2.5_lambda_aux_1

- WA: ep_160_milestone_4_lr_0.05_lr_decay_0.1_batch_64_w_decay_0.001_scheduler_cosine T_2_lambda_kd_3

- DER: ep_200_milestone_3_lr_0.2_lr_decay_0.1_batch_256_w_decay_0.001_scheduler_cosine lambda_aux_2

- BEEF: ep_200_milestone_3_lr_0.15_lr_decay_0.1_batch_128_w_decay_0.0001_scheduler_cosine fusion_ep_200_energy_w_0.02_logits_align_2.3

**Best hyperparameters (CIFAR-50, 6 Tasks)** The following represents the best hyperparameters of each algorithm selected in the hyperparameter tuning phase using CIFAR-50 (6 Tasks).

- Replay: ep_70_milestone_2_lr_0.05_lr_decay_0.1_batch_32_w_decay_0.0001_scheduler_cosine

- BiC: ep_120_milestone_2_lr_0.05_lr_decay_0.3_batch_32_w_decay_0.0001_scheduler_cosine T_2.5_lambda_kd_1.5_split_ratio_0.3

- PODNet: ep_30_milestone_3_lr_0.05_lr_decay_0.5_batch_64_w_decay_0.0005_scheduler_cosine lambda_c_1_lambda_f_3_nb_proxy_30_ft_epochs_50_ft_lrate_0.003_adaptive_factor_False

- FOSTER: ep_70_milestone_2_lr_0.05_lr_decay_0.1_batch_64_w_decay_0.0005_scheduler_steplr T_1.5_lambda_kd_1_fe_3_beta_0.97_0.93_comp_ep_200

- MEMO: ep_160_milestone_4_lr_0.05_lr_decay_0.1_batch_32_w_decay_0.001_scheduler_cosine lambda_aux_0.5_examplar_bs_256

- iCaRL: ep_120_milestone_2_lr_0.05_lr_decay_0.1_batch_32_w_decay_0.0005_scheduler_steplr T_1_lambda_aux_1

- WA: ep_160_milestone_3_lr_0.05_lr_decay_0.1_batch_64_w_decay_0.0001_scheduler_cosine T_2_lambda_kd_1.5

- DER: ep_120_milestone_3_lr_0.05_lr_decay_0.5_batch_64_w_decay_0.001_scheduler_cosine lambda_aux_1.5

- BEEF: ep_30_milestone_4_lr_0.05_lr_decay_0.1_batch_128_w_decay_0.0001_scheduler_steplr fusion_ep_70_energy_w_0.01_logits_align_1.4

**Best hyperparameters (ImageNet-50, 10 Tasks)** The following represents the best hyperparameters of each algorithm selected in the hyperparameter tuning phase using ImageNet-50 (10 Tasks).

- Replay: ep_70_milestone_3_lr_0.2_lr_decay_0.1_batch_64_w_decay_0.0001_scheduler_steplr

- BiC: ep_120_milestone_3_lr_0.1_lr_decay_0.1_batch_32_w_decay_0.0001_scheduler_steplr T_1_lambda_kd_3_split_ratio_0.1

- PODNet: ep_30_milestone_4_lr_0.2_lr_decay_0.3_batch_64_w_decay_0.0005_scheduler_cosine lambda_c_7_lambda_f_1_nb_proxy_50_ft_epochs_20_ft_lrate_0.007_adaptive_factor_True

- FOSTER: ep_120_milestone_3_lr_0.1_lr_decay_0.1_batch_32_w_decay_0.0001_scheduler_steplr T_1_lambda_kd_3_fe_1_beta_0.99_0.93_comp_ep_160

- MEMO: ep_120_milestone_3_lr_0.1_lr_decay_0.1_batch_32_w_decay_0.0001_scheduler_steplr lambda_aux_1_examplar_bs_256

- iCaRL: ep_200_milestone_3_lr_0.15_lr_decay_0.1_batch_64_w_decay_0.0001_scheduler_cosine T_2.5_lambda_aux_2

- WA: ep_200_milestone_3_lr_0.15_lr_decay_0.1_batch_64_w_decay_0.0001_scheduler_cosine T_2.5_lambda_kd_2

- DER: ep_200_milestone_3_lr_0.15_lr_decay_0.1_batch_64_w_decay_0.0001_scheduler_cosine lambda_aux_3

- BEEF NaN

**Best hyperparameters (ImageNet-50, 6 Tasks)** The following represents the best hyperparameters of each algorithm selected in the hyperparameter tuning phase using ImageNet-50 (6 Tasks).

- Replay: ep_200_milestone_2_lr_0.2_lr_decay_0.3_batch_32_w_decay_0.0001_scheduler_steplr

- BiC: ep_120_milestone_3_lr_0.1_lr_decay_0.1_batch_32_w_decay_0.0001_scheduler_steplr T_1_lambda_kd_3_split_ratio_0.1

- PODNet: ep_30_milestone_4_lr_0.2_lr_decay_0.3_batch_64_w_decay_0.0005_scheduler_cosine lambda_c_7_lambda_f_1_nb_proxy_50_ft_epochs_20_ft_lrate_0.007_adaptive_factor_True

- FOSTER: ep_30_milestone_4_lr_0.2_lr_decay_0.3_batch_64_w_decay_0.0005_scheduler_cosine T_2_lambda_kd_1_fe_2_beta_0.97_0.99_comp_ep_120

- MEMO: ep_120_milestone_3_lr_0.15_lr_decay_0.1_batch_512_w_decay_0.001_scheduler_steplr lambda_aux_0.5_examplar_bs_32

- iCaRL: ep_200_milestone_3_lr_0.15_lr_decay_0.1_batch_64_w_decay_0.0001_scheduler_cosine T_2.5_lambda_aux_2

- WA: ep_120_milestone_4_lr_0.1_lr_decay_0.5_batch_64_w_decay_0.0005_scheduler_steplr T_1_lambda_kd_1

- DER: ep_120_milestone_4_lr_0.3_lr_decay_0.5_batch_128_w_decay_0.0005_scheduler_cosine lambda_aux_0.5

- BEEF: NaN

### B.2 EXPERIMENTAL SETTINGS FOR CLASS-INCREMENTAL LEARNING WITH A PRETRAINED MODEL

**Experimental details**   For experiments using the proposed evaluation protocol on class-incremental learning algorithms with a pretrained model, we employ the PILOT (Sun et al., 2023) code for each algorithm. The experimental setup closely followed PILOT's environment, using Python 3.8, PyTorch 2.0.1, and CUDA 11.7.

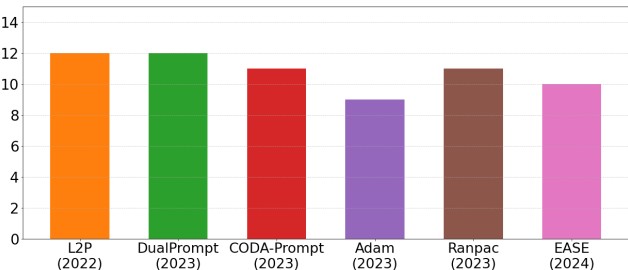

Figure 11: # of hyperparameters.

**Pretrained hyperparameters**   The process of selecting hyperparameters for algorithms using a pretrained model is similar to the previous experiments. We comprehensively consider both general hyperparameters and algorithm-specific ones, finding the best hyperparameters during the tuning phase. Figure 11 shows the number of hyperparameters for each algorithm. The predefined hyperparameters used for this process are listed in Table 5. Using the selected hyperparameters, we train each algorithm across the entire CL scenario. The range of each hyperparameter is set based on values reported in previous work for each type of algorithm. Unlike the algorithms without pretrained models, which use the same optimizer (*i.e.*, SGD), different optimizers have been used across algorithms in this case, so we also perform sampling for the optimizer. For hyperparameters of the optimizer that were not sampled, we use the default values provided in PyTorch.

Table 5: The predefined set of hyperparametes for class-IL with a pretrained model.

| Algorithm | Hyperparameter Name | $h^{Set}$ |
|---|---|---|
| All algorithms | Epoch | [3, 5, 10, 15, 20, 25]
(for L2P, DualPrompt. CODA-Pormpt)
/ [5, 10, 15, 20, 25, 30]
(for Adam-Adapter, Ranpac, EASE) |
| | LR | [0.000875, 0.001375, 0.001875, 0.002375, 0.0025]
(for L2P, DualPrompt. CODA-Pormpt)
/ [0.01, 0.02, 0.03, 0.04, 0.05]
(for Adam-Adapter, Ranpac, EASE) |
| | Num milestones | [2, 3, 4] |
| | LR decay | [0.1, 0.3, 0.5] |
| | Batch size | [8, 16, 24, 48, 64, 128]
(for L2P, DualPrompt, CODA-Prompt, Adam-Adapter |
| | Weigh decay | [0, 0.0001, 0.0005]
(for L2P, DualPrompt, CODA-Prompt)
/ [0.0001, 0.0005, 0.001, 0.005]
(for Adam-Adapter, Ranpac, EASE) |
| | LR Scheduler | ['steplr', 'cosine', 'constant'] |
| | Optimizer | ['sgd', 'adam', 'adamw'] |
| L2P, DualPrompt | M Size | [10, 15, 20, 25, 30] |
| L2P | Length ($L_p$) | [2, 4, 6, 8, 10] |
| L2P | Top k | [2, 4, 6, 8, 10] |
| L2P, DualPrompt | $\lambda$ | [0.1, 0.3, 0.5] |
| DualPrompt | Prompt length of g ($L_g$) | [5, 10, 15, 20, 30] |
| DualPrompt | Length ($L_e$) | [5, 10, 15, 20, 30] |
| CODA-Prompt | Pool size | [30, 50, 100, 200, 300] |
| CODA-Prompt | Prompt length | [4, 8, 16, 24, 32] |
| CODA-Prompt | Orthogonality Mu | [0.2, 0.1, 0.01, 0.001, 0] |
| Adam-Adapter, Ranpac, EASE | FFN num | [4,8,16,32,64] |
| Ranpac | M | [5000, 10000, 15000, 20000] |
| Ranpac | Prompt token num | [3, 5, 10, 20, 30, 50] |
| EASE | $\alpha$ | [0.01, 0.05, 0.1, 0.15, 0.2] |

**Original hyperparameters**  The following shows the original hyperparameters of each algorithm reported in PILOT.

- L2P_ep_10_milestone_3_lr_0.001875_lr_decay_0_batch_32_w_decay_0
  scheduler_constant_optimizer_adam_size_10_length_5_top_k_5_lamb_0.1

- DualPrompt_ep_10_milestone_4_lr_0.001_lr_decay_0.0_batch_24_w_decay_0.0
  scheduler_constant_optimizer_adam_size_10_L_e_5_L_g_5_top_k_1_lamb_0.1

- CODA-Prompt_ep_50_milestone_2_lr_0.001_lr_decay_0.0_batch_128_w_decay_0.0
  scheduler_cosine_optimizer_adam_e_pool_size_100_e_p_length_8_ortho_mu_0.0

- Adam_ep_10_milestone_3_lr_0.05_lr_decay_0.0_batch_16_w_decay_0.005
  scheduler_constant_optimizer_sgd_ffn_num_100

- Ranpac_ep_10_milestone_2_lr_0.05_lr_decay_0.0_batch_16_w_decay_0.005
  scheduler_constant_optimizer_sgd_ffn_num_64_M_10000_pt_num_30

- EASE_ep_20_milestone_4_lr_0.05_lr_decay_0.0_batch_16_w_decay_0.005
  scheduler_cosine_optimizer_sgd_ffn_num_64_alpha_0.1

**Best hyperparameters (CUB-200, 20 Tasks)**  The following represents the best hyperparameters of each algorithm selected in the hyperparameter tuning phase using CUB-200 (20 Tasks).

- L2P: ep_20_milestone_2_lr_0.002375_lr_decay_0.5_batch_64_w_decay_0.0001
  scheduler_constant_optimizer_adamw_size_15_length_6_top_k_4_lamb_0.1

- DualPrompt: ep_25_milestone_3_lr_0.000875_lr_decay_0.1_batch_48_w_decay_0.0005
  scheduler_constant_optimizer_adamw_size_20_L_e_5_L_g_30_top_k_1_lamb_0.3

- CODA-Prompt: ep_25_milestone_2_lr_0.000875_lr_decay_0.3_batch_24_w_decay_0.0005 scheduler_steplr_optimizer_sgd_e_pool_size_30_e_p_length_4_ortho_mu_0.01

- Adam: ep_15_milestone_4_lr_0.05_lr_decay_0.5_batch_48_w_decay_0.0005 scheduler_cosine_optimizer_sgd_ffn_num_8

- Ranpac: ep_30_milestone_4_lr_0.01_lr_decay_0.1_batch_8_w_decay_0.0005 scheduler_cosine_optimizer_sgd_ffn_num_32_M_20000_pt_num_5

- EASE: ep_15_milestone_4_lr_0.02_lr_decay_0.5_batch_128_w_decay_0.001 scheduler_cosine_optimizer_sgd_ffn_num_8_alpha_0.01

**Best hyperparameters (CUB-200, 10 Tasks)**  The following represents the best hyperparameters of each algorithm selected in the hyperparameter tuning phase using CUB-200 (10 Tasks).

- L2P: ep_25_milestone_2_lr_0.001875_lr_decay_0.3_batch_64_w_decay_0.0005 scheduler_cosine_optimizer_adamw_size_10_length_6_top_k_6_lamb_0.5

- DualPrompt: ep_25_milestone_3_lr_0.0025_lr_decay_0.5_batch_128_w_decay_0.0005 scheduler_steplr_optimizer_sgd_size_20_L_e_10_L_g_10_top_k_1_lamb_0.5

- CODA-Prompt: ep_25_milestone_3_lr_0.0025_lr_decay_0.3_batch_64_w_decay_0 scheduler_cosine_optimizer_adamw_e_pool_size_100_e_p_length_8_ortho_mu_0

- Adam: ep_20_milestone_3_lr_0.04_lr_decay_0.3_batch_8_w_decay_0.0005 scheduler_steplr_optimizer_sgd_ffn_num_32

- Ranpac: ep_30_milestone_4_lr_0.02_lr_decay_0.3_batch_16_w_decay_0.0001 scheduler_steplr_optimizer_sgd_ffn_num_64_M_10000_pt_num_3

- EASE: ep_15_milestone_4_lr_0.01_lr_decay_0.3_batch_64_w_decay_0.0005 scheduler_steplr_optimizer_sgd_ffn_num_8_alpha_0.05

**Best hyperparameters (ImageNet-R, 20 Tasks)**  The following represents the best hyperparameters of each algorithm selected in the hyperparameter tuning phase using ImageNet-R (20 Tasks).

- L2P_ep_25_milestone_3_lr_0.000875_lr_decay_0.5_batch_64_w_decay_0 scheduler_steplr_optimizer_adam_size_10_length_10_top_k_4_lamb_0.5

- DualPrompt: ep_15_milestone_4_lr_0.001875_lr_decay_0.5_batch_128_w_decay_0 scheduler_steplr_optimizer_adam_size_20_L_e_30_L_g_5_top_k_1_lamb_0.5

- CODA-Prompt: ep_15_milestone_2_lr_0.002375_lr_decay_0.1_batch_48_w_decay_0.0001 scheduler_cosine_optimizer_adamw_e_pool_size_300_e_p_length_32_ortho_mu_0.001

- Adam: ep_25_milestone_3_lr_0.05_lr_decay_0.5_batch_64_w_decay_0.001 scheduler_constant_optimizer_sgd_ffn_num_64

- Ranpac: ep_20_milestone_2_lr_0.05_lr_decay_0.3_batch_24_w_decay_0.0005 scheduler_constant_optimizer_sgd_ffn_num_16_M_15000_pt_num_20

- EASE: ep_15_milestone_4_lr_0.04_lr_decay_0.1_batch_24_w_decay_0.0001 scheduler_constant_optimizer_sgd_ffn_num_16_alpha_0.15

**Best hyperparameters (ImageNet-R, 10 Tasks)**  The following represents the best hyperparameters of each algorithm selected in the hyperparameter tuning phase using ImageNet-R (10 Tasks).

- L2P: ep_25_milestone_3_lr_0.001375_lr_decay_0.5_batch_128_w_decay_0 scheduler_constant_optimizer_adamw_size_20_length_6_top_k_10_lamb_0.3

- DualPrompt: ep_25_milestone_2_lr_0.001375_lr_decay_0.3_batch_128_w_decay_0.0005 scheduler_constant_optimizer_adamw_size_30_L_e_20_L_g_20_top_k_1_lamb_0.3

- CODA-Prompt: ep_20_milestone_2_lr_0.001375_lr_decay_0.1_batch_48_w_decay_0 scheduler_steplr_optimizer_adam_e_pool_size_300_e_p_length_8_ortho_mu_0

- Adam: ep_30_milestone_2_lr_0.05_lr_decay_0.1_batch_64_w_decay_0.001 scheduler_cosine_optimizer_sgd_ffn_num_32

- Ranpac: ep_20_milestone_3_lr_0.03_lr_decay_0.1_batch_24_w_decay_0.0001 scheduler_steplr_optimizer_sgd_ffn_num_64_M_20000_pt_num_20

- EASE: ep_30_milestone_4_lr_0.05_lr_decay_0.3_batch_128_w_decay_0.001 scheduler_cosine_optimizer_adam_ffn_num_16

**Best hyperparameters (CUB-100-1, 20 Tasks)** The following represents the best hyperparameters of each algorithm selected in the hyperparameter tuning phase using CUB-100-1 (20 Tasks).

- L2P: ep_20_milestone_3_lr_0.002375_lr_decay_0.3_batch_128_w_decay_0.0005 scheduler_constant_optimizer_adamw_size_20_length_8_top_k_4_lamb_0.5
- DualPrompt: ep_25_milestone_4_lr_0.001375_lr_decay_0.1_batch_128_w_decay_0 scheduler_constant_optimizer_adam_size_15_L_e_15_L_g_20_top_k_1_lamb_0.5
- CODA-Prompt: ep_10_milestone_4_lr_0.0025_lr_decay_0.3_batch_64_w_decay_0 scheduler_constant_optimizer_adam_e_pool_size_200_e_p_length_4_ortho_mu_0.001
- Adam: ep_5_milestone_2_lr_0.05_lr_decay_0.5_batch_16_w_decay_0.0005 scheduler_constant_optimizer_sgd_ffn_num_4
- Ranpac: ep_15_milestone_4_lr_0.04_lr_decay_0.1_batch_128_w_decay_0.0001 scheduler_cosine_optimizer_sgd_ffn_num_4_M_15000_pt_num_30
- EASE: ep_10_milestone_4_lr_0.01_lr_decay_0.1_batch_16_w_decay_0.001 scheduler_constant_optimizer_sgd_ffn_num_32_alpha_0.05

**Best hyperparameters (CUB-100-1, 10 Tasks)** The following represents the best hyperparameters of each algorithm selected in the hyperparameter tuning phase using CUB-100-1 (10 Tasks).

- L2P: ep_25_milestone_2_lr_0.0025_lr_decay_0.3_batch_128_w_decay_0 scheduler_cosine_optimizer_adam_size_20_length_4_top_k_6_lamb_0.5
- DualPrompt: ep_25_milestone_4_lr_0.002375_lr_decay_0.3_batch_128_w_decay_0.0005 scheduler_cosine_optimizer_adamw_size_15_L_e_30_L_g_15_top_k_1_lamb_0.5
- CODA-Prompt: ep_25_milestone_3_lr_0.001375_lr_decay_0.1_batch_64_w_decay_0.0001 scheduler_cosine_optimizer_adamw_e_pool_size_50_e_p_length_4_ortho_mu_0
- Adam: ep_25_milestone_2_lr_0.05_lr_decay_0.3_batch_24_w_decay_0.0001 scheduler_steplr_optimizer_sgd_ffn_num_32
- Ranpac: ep_25_milestone_3_lr_0.02_lr_decay_0.3_batch_16_w_decay_0.0005 scheduler_constant_optimizer_sgd_ffn_num_16_M_10000_pt_num_20
- EASE: ep_15_milestone_3_lr_0.03_lr_decay_0.5_batch_128_w_decay_0.0001 scheduler_constant_optimizer_sgd_ffn_num_64_alpha_0.05

**Best hyperparameters (ImageNet-R-1, 20 Tasks)** The following represents the best hyperparameters of each algorithm selected in the hyperparameter tuning phase using ImageNet-R-1 (20 Tasks).

- L2P: ep_15_milestone_2_lr_0.002375_lr_decay_0.5_batch_48_w_decay_0 scheduler_cosine_optimizer_adamw_size_25_length_6_top_k_10_lamb_0.3
- DualPrompt: ep_20_milestone_3_lr_0.001875_lr_decay_0.1_batch_128_w_decay_0 scheduler_steplr_optimizer_adam_size_10_L_e_30_L_g_30_top_k_1_lamb_0.1
- CODA-Prompt: ep_15_milestone_2_lr_0.002375_lr_decay_0.5_batch_64_w_decay_0.0001 scheduler_cosine_optimizer_adamw_e_pool_size_100_e_p_length_4_ortho_mu_0.01
- Adam: ep_25_milestone_3_lr_0.04_lr_decay_0.3_batch_24_w_decay_0.001 scheduler_constant_optimizer_sgd_ffn_num_16
- Ranpac: ep_10_milestone_2_lr_0.02_lr_decay_0.3_batch_8_w_decay_0.0001 scheduler_cosine_optimizer_sgd_ffn_num_8_M_20000_pt_num_10
- EASE: ep_15_milestone_4_lr_0.03_lr_decay_0.5_batch_16_w_decay_0.0005 scheduler_constant_optimizer_sgd_ffn_num_64_alpha_0.05

**Best hyperparameters (ImageNet-R-1, 10 Tasks)** The following represents the best hyperparameters of each algorithm selected in the hyperparameter tuning phase using ImageNet-R-1 (10 Tasks).

- L2P: ep_20_milestone_3_lr_0.000875_lr_decay_0.1_batch_24_w_decay_0.0001 scheduler_cosine_optimizer_adamw_size_20_length_8_top_k_10_lamb_0.5

- DualPrompt: ep_25_milestone_4_lr_0.000875_lr_decay_0.3_batch_64_w_decay_0 scheduler_cosine_optimizer_adam_size_15_L_e_30_L_g_20_top_k_1_lamb_0.1

- CODA-Prompt: ep_15_milestone_3_lr_0.001375_lr_decay_0.5_batch_64_w_decay_0.0005 scheduler_constant_optimizer_adamw_e_pool_size_300_e_p_length_4_ortho_mu_0.01

- Adam: ep_25_milestone_3_lr_0.04_lr_decay_0.3_batch_24_w_decay_0.001 scheduler_constant_optimizer_sgd_ffn_num_16

- Ranpac: ep_25_milestone_4_lr_0.05_lr_decay_0.1_batch_24_w_decay_0.0001 scheduler_constant_optimizer_sgd_ffn_num_64_M_20000_pt_num_10

- EASE: ep_10_milestone_4_lr_0.04_lr_decay_0.1_batch_24_w_decay_0.001 scheduler_constant_optimizer_sgd_ffn_num_64_alpha_0.2

# C    ADDITIONAL EXPERIMENTAL RESULTS ON THE EVALUATION PHASE

## C.1    RESULT TABLES

**Class-IL without a pretrained model ($D^{HT}$ = ImageNet-100-1)**

Table 6: The experimental results of class-IL without a pretrained model (using original hyperparameters) The values in parentheses represent the standard deviation.

| 10 Tasks (Acc / AvgAcc) | $D^{HT}$ = ImageNet-100 |
|---|---|
| Replay | 41.21(1.06) / 59.82(1.48) |
| iCaRL | 40.50(1.19) / 60.12(1.41) |
| BiC | 39.61(2.39) / 64.27(1.59) |
| WA | 53.34(1.39) / 68.92(1.54) |
| PODNet | 46.66(1.11) / 64.13(1.20) |
| DER | 61.96(1.04) / 72.10(1.41) |
| FOSTER | 60.68(0.71) / 69.97(1.70) |
| BEEF | NaN |
| MEMO | 59.59(1.29) / 70.04(1.62) |

Table 7: The experimental results of class-IL without a pretrained model (using $D^{HT}$ = ImageNet-100-1) in the hyperparameter tuning phase. The values in parentheses represent the standard deviation.

| 10 Tasks (Acc / AvgAcc) | $D^{HT}$ = ImageNet-100-1 | $D^E$ = ImageNet-100-2 |
|---|---|---|
| Replay | 44.78(1.19) / 59.85(0.95) | 44.27(1.05) / 61.49(0.87) |
| iCaRL | 42.58(1.06) / 61.27(1.26) | 42.44(1.50) / 63.39(1.18) |
| BiC | 54.22(1.27) / 67.31(0.74) | 58.77(0.96) / 71.81(1.42) |
| WA | 54.67(0.60) / 69.54(1.41) | 59.89(1.18) / 72.93(1.94) |
| PODNet | 55.35(0.93) / 68.74(1.52) | 57.48(0.94) / 71.76(1.62) |
| DER | 63.31(0.42) / 72.93(0.87) | 70.23(0.46) / 77.12(1.20) |
| FOSTER | 58.36(0.85) / 71.99(0.98) | 61.46(0.98) / 68.41(1.23) |
| BEEF | NaN | NaN |
| MEMO | 57.91(0.54) / 71.25(1.41) | 61.94(0.78) / 71.35(2.17) |

Table 8: The experimental results of class-IL without a pretrained model (using $D^{HT}$ = ImageNet-100-1) in the hyperparameter tuning phase. The values in parentheses represent the standard deviation.

| 6 Tasks (Acc / AvgAcc) | $D^{HT}$ = ImageNet-100 | $D^E$ = ImageNet-200 |
|---|---|---|
| Replay | 42.93(2.41) / 53.81(1.72) | 43.26(1.38) / 49.28(0.53) |
| iCaRL | 46.62(1.54) / 57.27(0.73) | 45.64(1.49) / 59.18(0.54) |
| BiC | 37.14(1.62) / 36.42(1.89) | 38.43(2.53) / 40.89(3.07) |
| WA | 58.72(1.02) / 65.58(1.55) | 60.58(1.35) / 69.47(1.71) |
| PODNet | 67.22(0.67) / 75.05(1.16) | 65.51(1.83) / 75.82(1.03) |
| DER | 72.20(0.51) / 77.68(1.08) | 75.83(0.64) / 81.19(0.70) |
| FOSTER | 69.48(0.50) / 74.59(1.18) | 71.62(1.08) / 78.29(1.14) |
| BEEF | 74.67(0.14) / 78.92(0.54) | 75.09(0.29) / 81.31(0.50) |
| MEMO | 59.91(0.87) / 67.22(1.63) | 62.80(3.16) / 68.77(6.26) |

**Class-IL without a pretrained model ($D^{HT}$ = CIFAR-50-1)**

Table 9: The experimental results of class-IL without a pretrained model (using $D^{HT}$ = CIFAR-50-1) in the hyperparameter tuning phase.) The values in parentheses represent the standard deviation.

| 10 Tasks (Acc / AvgAcc) | $D^E$ = CIFAR-50-2 | $D^E$ = ImageNet-50-2 |
|---|---|---|
| Replay | 45.42(2.19) / 65.88(1.97) | 42.51(0.47) / 60.72(1.58) |
| iCaRL | 47.12(2.80) / 66.71(2.07) | 42.44(1.00) / 61.55(1.64) |
| BiC | 52.83(2.83) / 69.16(2.30) | 49.52(1.16) / 67.09(1.74) |
| WA | 54.89(2.13) / 69.85(2.32) | 53.64(1.47) / 67.75(1.90) |
| PODNet | 51.20(1.76) / 69.47(0.13) | 51.70(1.19) / 67.86(1.67) |
| DER | 63.51(1.98) / 75.04(1.24) | 63.40(1.02) / 72.67(1.62) |
| FOSTER | 60.00(2.72) / 72.29(2.09) | 62.09(1.83) / 70.24(1.50) |
| BEEF | 57.24(1.48) / 72.26(2.05) | NaN |
| MEMO | 60.72(2.41) / 73.78(1.99) | 54.91(1.59) / 68.06(2.10) |

Table 10: The experimental results of class-IL without a pretrained model (using $D^{HT}$ = CIFAR-50-1) in the hyperparameter tuning phase.) The values in parentheses represent the standard deviation.

| 6 Tasks (Acc / AvgAcc) | $D^E$ = CIFAR-50-2 | $D^E$ = ImageNet-50-2 |
|---|---|---|
| Replay | 48.00(1.98) / 59.86(1.03) | 46.30(1.31) / 55.67(0.64) |
| iCaRL | 46.09(1.51) / 59.14(1.39) | 46.21(1.72) / 57.79(1.06) |
| BiC | 58.22(1.20) / 68.16(1.96) | 46.26(3.26) / 59.07(3.87) |
| WA | 61.37(1.02) / 70.56(0.51) | 61.47(0.72) / 69.67(0.63) |
| PODNet | 62.62(0.39) / 72.62(0.75) | 64.30(0.78) / 73.56(1.01) |
| DER | 67.98(1.34) / 75.88(0.78) | 70.68(0.75) / 76.56(0.95) |
| FOSTER | 66.45(0.55) / 73.93(0.77) | 69.86(0.45) / 75.27(0.83) |
| BEEF | 65.51(1.29) / 72.98(0.50) | NaN |
| MEMO | 64.64(1.54) / 73.50(0.83) | 51.40(3.39) / 62.11(3.33) |

**Class-IL without a pretrained model ($D^{HT}$ = ImageNet-50-1)**

Table 11: The experimental results of class-IL without a pretrained model (using $D^{HT}$ = ImageNet-50-1) in the hyperparameter tuning phase.) The values in parentheses represent the standard deviation.

| 10 Tasks (Acc / AvgAcc) | $D^E$ = ImageNet-50-2 | $D^E$ = CIFAR-50-2 |
|---|---|---|
| Replay | 43.71(0.81) / 58.75(1.60) | 44.19(2.17) / 63.57(1.50) |
| iCaRL | 39.41(1.46) / 59.51(1.70) | 41.59(3.10) / 62.42(2.85) |
| BiC | 51.26(1.39) / 65.33(2.48) | 51.22(3.67) / 66.41(2.92) |
| WA | 51.85(0.79) / 67.23(1.79) | 57.72(1.92) / 71.39(2.00) |
| PODNet | 51.31(1.24) / 67.28(1.53) | 48.19(1.17) / 65.77(1.29) |
| DER | 64.89(1.16) / 74.15(1.56) | 63.64(1.32) / 75.32(1.21) |
| FOSTER | 61.57(0.70) / 72.38(1.20) | 58.64(2.15) / 72.89(1.81) |
| BEEF | NaN | NaN |
| MEMO | 57.56(1.24) / 68.36(2.27) | 58.99(1.01) / 72.43(1.81) |

Table 12: The experimental results of class-IL without a pretrained model (using $D^{HT}$ = ImageNet-50-1) in the hyperparameter tuning phase.) The values in parentheses represent the standard deviation.

| 6 Tasks (Acc / AvgAcc) | $D^E$ = ImageNet-50-2 | $D^E$ = CIFAR-50-2 |
|---|---|---|
| Replay | 42.82(1.43) / 53.50(1.54) | 42.28(0.71) / 52.18(1.31) |
| iCaRL | 42.47(1.73) / 54.65(1.85) | 40.24(2.64) / 52.89(2.14) |
| BiC | 44.68(2.81) / 54.19(2.93) | 39.65(1.32) / 49.49(1.46) |
| WA | 55.68(0.07) / 64.69(0.72) | 56.14(1.99) / 64.08(1.60) |
| PODNet | 64.10(0.80) / 72.50(0.81) | 61.33(0.54) / 71.27(1.07) |
| DER | 70.28(0.98) / 76.14(1.00) | 64.76(1.06) / 72.89(1.28) |
| FOSTER | 68.40(1.08) / 75.02(0.94) | 65.31(0.26) / 73.80(0.68) |
| BEEF | NaN | NaN |
| MEMO | 50.92(1.25) / 60.93(1.67) | 50.58(2.62) / 60.66(2.65) |

**Class-IL with a pretrained model ($D^{HT}$ = CUB-200)**

Table 13: The experimental results of class-IL with a pretrained model (using original hyperparameters) The values in parentheses represent the standard deviation.

| 10 Tasks (Acc / AvgAcc) | $D^{HT}$ = CUB-200 |
|---|---|
| L2P | 72.32(0.62) / 76.82(0.30) |
| DualPrompt | 68.74(0.54) / 74.39(0.68) |
| CODA-Prompt | 75.19(0.33) / 80.27(0.93) |
| Adam | 71.21(1.06) / 77.52(1.24) |
| Ranpac | 78.27(0.57) / 83.24(0.44) |
| EASE | 77.07(0.19) / 82.65(0.68) |

Table 14: The experimental results of class-IL with a pretrained model (using $D^{HT}$ = CUB-200) in the hyperparameter tuning phase. The values in parentheses represent the standard deviation.

| 20 Tasks (Acc / AvgAcc) | $D^E$ = ImageNet-R | $D^E$ = ImageNet-A |
|---|---|---|
| L2P | 69.93(0.39) / 75.90(0.23) | 40.92(1.53) / 51.24(1.39) |
| DualPrompt | 67.20(0.78) / 73.79(0.64) | 44.00(1.07) / 54.12(0.96) |
| CODA-Prompt | 68.63(0.64) / 74.61(0.84) | 48.20(1.05) / 57.94(0.87) |
| Adam | 67.70(1.38) / 74.45(1.35) | 49.61(0.29) / 59.67(0.80) |
| Ranpac | 78.72(0.40) / 83.71(0.56) | 62.95(1.41) / 68.64(2.58) |
| EASE | 61.94(0.06) / 68.36(0.63) | 49.37(0.12) / 59.48(0.75) |

Table 15: The experimental results of class-IL with a pretrained model (using $D^{HT}$ = CUB-200) in the hyperparameter tuning phase. The values in parentheses represent the standard deviation.

| 10 Tasks (Acc / AvgAcc) | $D^E$ = ImageNet-R | $D^E$ = ImageNet-A |
|---|---|---|
| L2P | 71.86(0.66) / 77.42(0.92) | 45.13(1.25) / 53.57(0.92) |
| DualPrompt | 66.33(0.42) / 73.03(0.60) | 39.97(2.32) / 52.58(0.70) |
| CODA-Prompt | 72.86(0.44) / 78.49(0.99) | 51.63(0.50) / 61.00(0.47) |
| Adam | 72.68(0.77) / 79.09(0.89) | 57.03(0.47) / 66.50(1.22) |
| Ranpac | 79.59(0.29) / 84.46(0.41) | 66.14(0.40) / 73.63(1.05) |
| EASE | 61.96(0.06) / 67.74(0.67) | 49.32(0.48) / 58.30(0.86) |

**Class-IL with a pretrained model ($D^{HT}$ = ImageNet-R)**

Table 16: The experimental results of class-IL with a pretrained model (using $D^{HT}$ = ImageNet-R) in the hyperparameter tuning phase. The values in parentheses represent the standard deviation.

| 20 Tasks (Acc / AvgAcc) | $D^E$ = CUB-200 | $D^E$ = ImageNet-A |
|---|---|---|
| L2P | 63.76(1.81) / 76.59(1.48) | 36.97(1.31) / 46.78(0.71) |
| DualPrompt | 68.78(0.78) / 79.67(1.04) | 47.54(0.79) / 55.91(0.84) |
| CODA-Prompt | 67.92(2.11) / 79.65(1.93) | 50.07(0.29) / 59.76(0.58) |
| Adam | 85.38(0.19) / 90.87(0.90) | 53.86(1.44) / 63.99(2.61) |
| Ranpac | 89.86(0.22) / 93.44(0.78) | 38.53(31.11) / 67.65(3.37) |
| EASE | 79.89(1.22) / 87.58(1.19) | 53.99(1.05) / 64.11(0.78) |

Table 17: The experimental results of class-IL with a pretrained model (using $D^{HT}$ = ImageNet-R) in the hyperparameter tuning phase. The values in parentheses represent the standard deviation.

| 10 Tasks (Acc / AvgAcc) | $D^E$ = CUB-200 | $D^E$ = ImageNet-A |
|---|---|---|
| L2P | 69.75(1.79) / 79.92(1.24) | 43.50(0.99) / 50.06(1.18) |
| DualPrompt | 71.74(1.01) / 82.22(1.10) | 39.47(0.79) / 50.63(0.94) |
| CODA-Prompt | 72.30(1.11) / 83.00(1.35) | 52.39(0.38) / 61.87(1.01) |
| Adam | 85.90(0.17) / 90.93(0.89) | 56.63(0.78) / 65.94(1.45) |
| Ranpac | 89.99(0.29) / 93.36(0.83) | 63.78(1.52) / 71.70(1.88) |
| EASE | 74.00(0.78) / 83.69(0.74) | 54.76(1.36) / 66.14(1.65) |

**Class-IL with a pretrained model ($D^{HT}$ = CUB-100-1)**

Table 18: The experimental results of class-IL with a pretrained model (using $D^{HT}$ = CUB-100-1) in the hyperparameter tuning phase. The values in parentheses represent the standard deviation.

| 20 Tasks (Acc / AvgAcc) | $D^E$ = CUB-100-2 | $D^E$ = ImageNet-R-2 | ImageNet-A-2 |
|---|---|---|---|
| L2P | 54.12(3.59) / 68.33(3.73) | 66.01(0.74) / 72.17(1.04) | 28.08(2.38) / 39.18(2.75) |
| DualPrompt | 59.83(1.63) / 73.54(2.68) | 65.51(0.32) / 71.58(0.68) | 33.90(2.26) / 44.84(2.25) |
| CODA-Prompt | 58.16(1.88) / 71.05(2.68) | 66.73(0.61) / 73.06(0.46) | 30.62(0.82) / 41.70(1.70) |
| Adam | 85.95(0.08) / 90.56(0.24) | 67.77(0.84) / 74.53(1.74) | 43.93(0.09) / 55.63(2.69) |
| Ranpac | 89.52(0.35) / 90.52(2.96) | 74.53(0.28) / 79.80(0.81) | 30.30(22.41) / 45.87(4.57) |
| EASE | 85.19(0.49) / 89.91(0.74) | 67.17(0.29) / 73.61(0.75) | 44.11(0.29) / 55.42(2.83) |

Table 19: The experimental results of class-IL with a pretrained model (using $D^{HT}$ = CUB-100-1) in the hyperparameter tuning phase. The values in parentheses represent the standard deviation.

| 10 Tasks (Acc / AvgAcc) | $D^E$ = CUB-100-2 | $D^E$ = ImageNet-R-2 | ImageNet-A-2 |
|---|---|---|---|
| L2P | 66.15(1.41) / 76.68(1.49) | 70.11(0.53) / 75.61(0.87) | 34.96(0.92) / 44.98(2.26) |
| DualPrompt | 67.20(2.59) / 78.28(1.68) | 68.29(0.49) / 74.32(0.89) | 38.43(1.52) / 49.15(2.43) |
| CODA-Prompt | 68.37(2.71) / 78.93(2.57) | 70.35(0.81) / 75.59(0.90) | 37.23(1.87) / 47.48(1.85) |
| Adam | 86.76(0.21) / 90.75(0.46) | 72.73(0.27) / 79.42(0.59) | 44.81(0.85) / 55.08(2.22) |
| Ranpac | 90.60(0.36) / 93.08(0.65) | 80.40(0.3) / 85.00(0.47) | 49.56(2.52) / 57.60(1.96) |
| EASE | 85.86(0.10) / 90.11(0.26) | 63.36(0.03) / 69.36(0.95) | 43.88(0.15) / 54.49(2.64) |

**Class-IL with a pretrained model ($D^{HT}$ = ImageNet-R-1)**

Table 20: The experimental results of class-IL with a pretrained model (using $D^{HT}$ = ImageNet-R-1) in the hyperparameter tuning phase. The values in parentheses represent the standard deviation.

| 20 Tasks (Acc / AvgAcc) | $D^E$ = ImageNet-R-2 | $D^E$ = CUB-100-2 | ImageNet-A-2 |
|---|---|---|---|
| L2P | 66.15(0.85) / 71.93(1.13) | 51.04(1.45) / 66.04(1.71) | 25.13(2.27) / 34.21(2.51) |
| DualPrompt | 65.77(0.78) / 71.83(1.17) | 57.13(3.40) / 71.15(2.25) | 31.96(2.49) / 41.71(1.76) |
| CODA-Prompt | 66.44(0.66) / 72.62(0.36) | 57.24(1.90) / 71.27(1.95) | 30.48(1.62) / 41.30(2.56) |
| Adam | 70.69(0.73) / 77.86(0.51) | 86.35(0.14) / 90.83(0.56) | 44.25(0.86) / 55.84(2.75) |
| Ranpac | 76.15(0.93) / 81.68(0.94) | 73.73(31.52) / 89.58(2.03) | 35.06(15.86) / 47.04(6.07) |
| EASE | 75.16(0.68) / 81.68(0.71) | 76.36(2.61) / 84.35(2.55) | 42.49(1.76) / 54.40(3.21) |

Table 21: The experimental results of class-IL with a pretrained model (using $D^{HT}$ = ImageNet-R-1) in the hyperparameter tuning phase. The values in parentheses represent the standard deviation.

| 10 Tasks (Acc / AvgAcc) | $D^E$ = ImageNet-R-2 | $D^E$ = CUB-100-2 | ImageNet-A-2 |
|---|---|---|---|
| L2P | 70.35(0.64) / 75.66(0.30) | 63.71(2.33) / 74.62(1.61) | 29.10(1.24) / 38.80(1.44) |
| DualPrompt | 69.97(0.25) / 75.93(0.62) | 66.66(1.12) / 78.11(1.43) | 32.42(0.68) / 42.31(2.02) |
| CODA-Prompt | 72.17(0.46) / 77.80(0.50) | 66.98(1.3) / 78.70(0.98) | 37.04(1.49) / 46.47(2.45) |
| Adam | 72.84(0.67) / 79.69(0.86) | 85.26(0.41) / 89.77(0.45) | 37.36(2.72) / 48.62(4.07) |
| Ranpac | 80.70(0.50) / 85.28(0.46) | 91.09(0.51) / 91.63(3.51) | 41.98(19.61) / 58.79(4.70) |
| EASE | 78.33(0.41) / 83.82(0.71) | 79.70(1.47) / 86.23(1.59) | 42.49(0.69) / 53.69(2.61) |

## C.2 TRAINING GRAPHS

**Class-IL without a pretrained model** ($D^{HT}$ = **CIFAR50-1**, $D^E$ = **CUB50-2**)

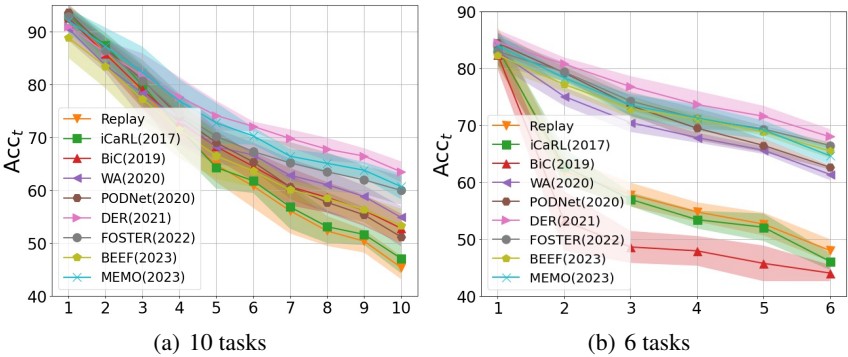

(a) 10 tasks
(b) 6 tasks

Figure 12: Experimental results on the evaluation phase.

**Class-IL without a pretrained model** ($D^{HT}$ = **CIFAR50-1**, $D^E$ = **ImageNet50-2**)

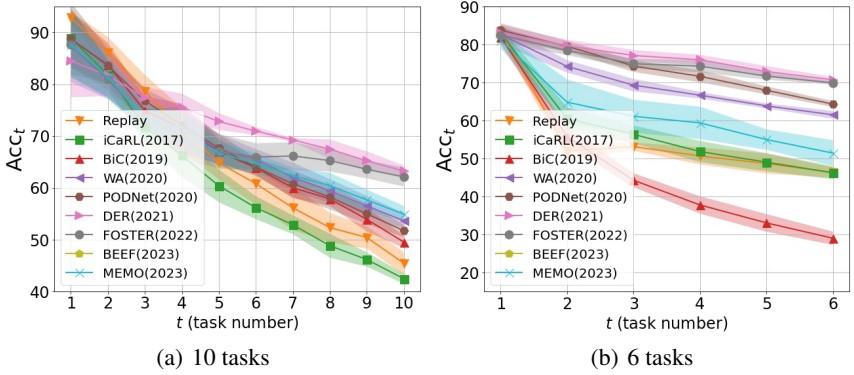

(a) 10 tasks
(b) 6 tasks

Figure 13: Experimental results on the evaluation phase.

**Class-IL without a pretrained model** ($D^{HT}$ = **ImageNet50-1**, $D^E$ = **ImageNet50-2**)

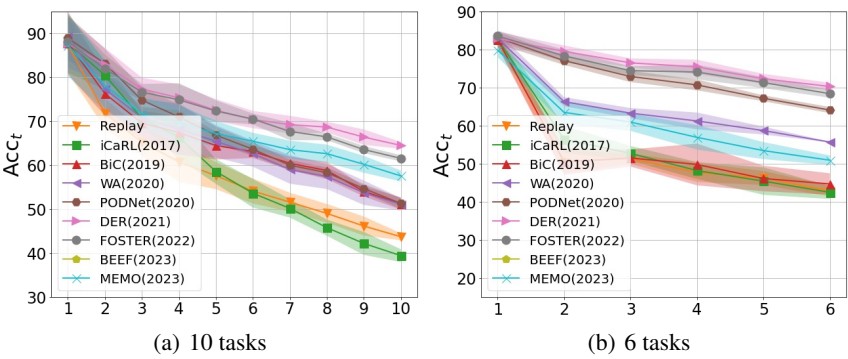

(a) 10 tasks
(b) 6 tasks

Figure 14: Experimental results on the evaluation phase.

**Class-IL without a pretrained model ($D^{HT} =$ ImageNet50-1, $D^E =$ CIFAR50-2)**

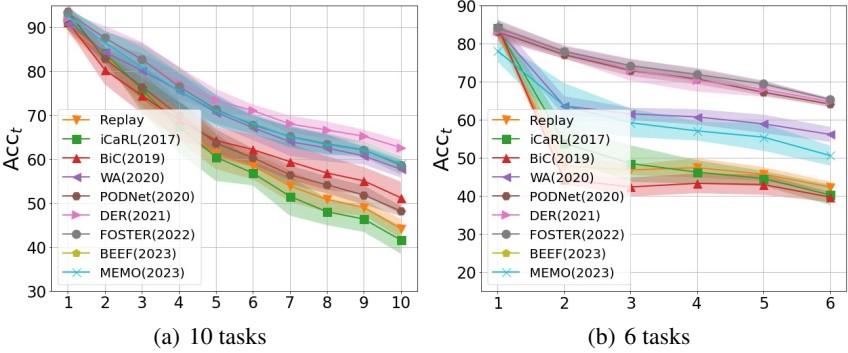

(a) 10 tasks

(b) 6 tasks

Figure 15: Experimental results on the evaluation phase.

**Class-IL with a pretrained model ($D^{HT} =$ CUB100-1, $D^E =$ CUB100-2)**

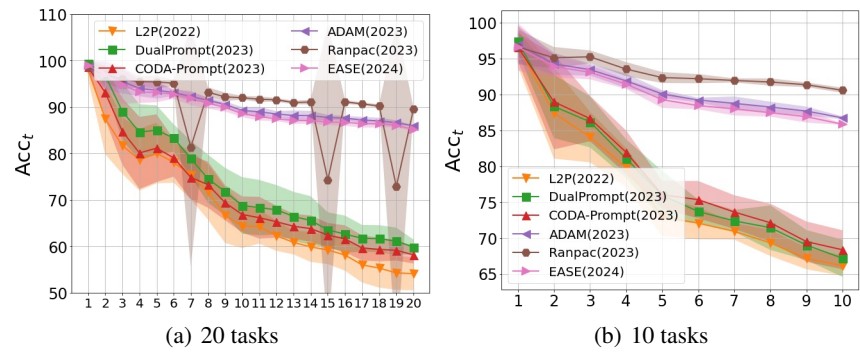

(a) 20 tasks

(b) 10 tasks

Figure 16: Experimental results on the evaluation phase.

**Class-IL with a pretrained model ($D^{HT} =$ CUB100-1, $D^E =$ ImageNet-R-2)**

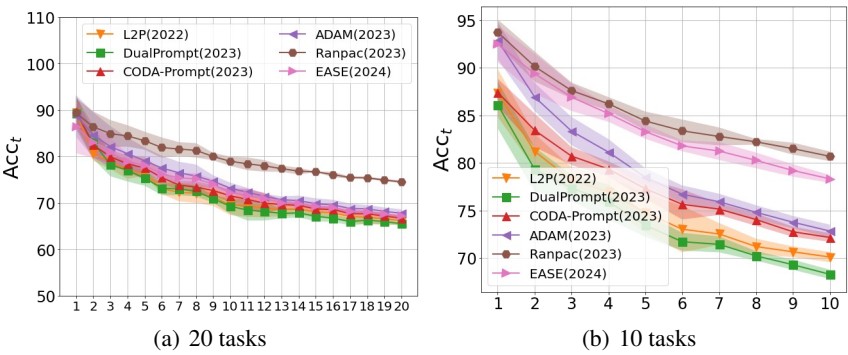

(a) 20 tasks

(b) 10 tasks

Figure 17: Experimental results on the evaluation phase.

**Class-IL with a pretrained model ($D^{HT} = $ ImageNet-R-1, $D^E = $ ImageNet-R-2)**

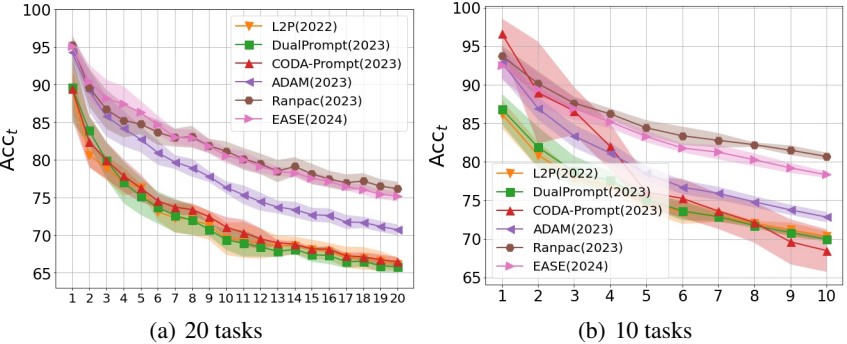

(a) 20 tasks      (b) 10 tasks

Figure 18: Experimental results on the evaluation phase.

**Class-IL with a pretrained model ($D^{HT} = $ ImageNet-R-1, $D^E = $ CUB100-2)**

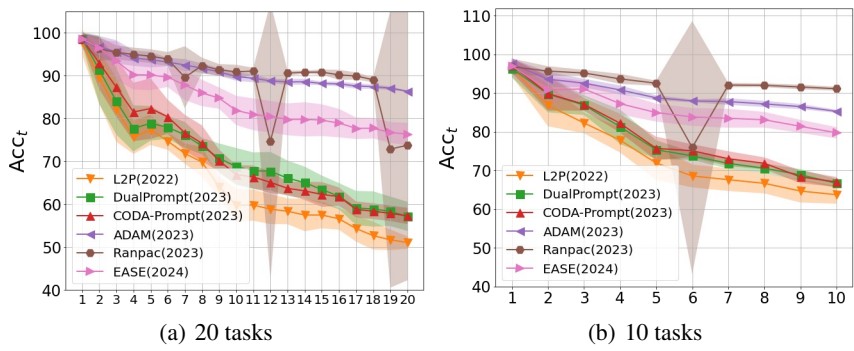

(a) 20 tasks      (b) 10 tasks

Figure 19: Experimental results on the evaluation phase.

**Class-IL with a pretrained model ($D^{HT} = $ ImageNet-R-1, $D^E = $ ImageNet-A-2)**

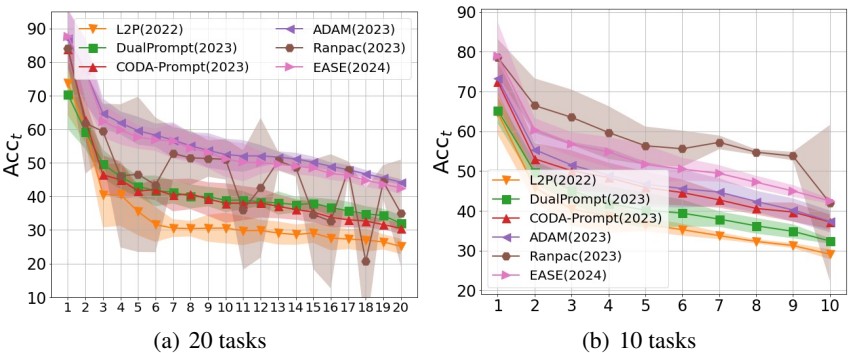

(a) 20 tasks      (b) 10 tasks

Figure 20: Experimental results on the evaluation phase.

## D LIMITATIONS AND FUTURE WORK

Our study has limitations. First, evaluating each algorithm using the proposed evaluation protocol requires a substantial number of training trials. Although we believe that our protocol serves as a basic method for more accurately assessing CL algorithms, it is not a perfect evaluation protocol. Consequently, developing more efficient protocols for accurately evaluating CL algorithms remains a significant and interesting research direction.

Second, we did not account for unpredictable CL scenarios, such as varying task numbers or class distributions. Our study assumes CL scenarios are predictable, but each phase's dataset differs. This is because, in real-world situations, some level of predictability is possible, and evaluating algorithms in completely unpredictable scenarios would be too harsh. Nevertheless, we believe that it is essential to explore evaluation methods for unpredictable scenarios in broader CL research, potentially through adaptive algorithms that can adjust hyperparameters for each task.

Finally, our evaluation focused solely on offline class-incremental learning algorithms. We think that similar challenges associated with the conventional evaluation protocol also exist in other CL domains, such as online class-incremental learning, class-incremental semantic segmentation, continual self-supervised learning, and continual reinforcement learning. As part of our future work, we intend to first apply the proposed protocol to online class-incremental learning algorithms, followed by its implementation in other domains.

