# OpenReview forum: "Hyperparameters in Continual Learning: A Reality Check"
_ICLR.cc/2025/Conference — ICLR 2025 Conference Withdrawn Submission_

### Official Review · Reviewer_irJH · 2024-10-17

**Soundness:** 2
**Presentation:** 2
**Contribution:** 2
**Rating:** 5
**Confidence:** 5

**Summary:**

This paper argued that the commonly used protocol of selecting hyperparameters for continual learning, which often select the best hyperparameter values within a given scenario and then evaluate the continual learning methods with these hyperparameters within the same scenario, is impractical in real-world applications. The authors then proposed an evaluation protocol consisting of a hyperparameter tuning phase and an evaluation phase on different datasets. The authors reported the performance variance of representative methods with this new protocol.

**Strengths:**

1. I appreciate the claim that the commonly used protocol of selecting hyperparameters for continual learning methods may not be optimal in applications, given that the old training samples are largely inaccessible.

2. The authors perform extensive experiments with a variety of continual learning methods under the proposed evaluation protocol.

**Weaknesses:**

This paper is essentially based on intuitive ideas and the empirical results are not very clear. It fails to cover many critical considerations in real-world applications.

1. The authors highlighted for many times that the two phases share the same scenario configuration (e.g., number of tasks) but are generated from different datasets. However, this consideration cannot fully reflect the possible differences across continual learning tasks, such as imbalanced classes per task, imbalanced training samples per class, blurred task boundaries, different task types, etc.

2. The experiments only consider class-incremental learning, rather than other typical scenarios such as task-incremental learning and domain-incremental learning.

3. Although continual learning methods show some performance differences between the two phases, most of them have similar trends (Figures 4 and 8). This reduces the significance of the proposed protocol, since the advanced methods exhibit consistent advantages.

4. The authors further analyzed the training cost. I agree that the training cost is a critical issue for continual learning, but it is almost orthogonal to the hyperparameter issue and independent of the proposed evaluation protocol.

**Questions:**

My major concerns lie in the coverage of hyperparameter issues in real-world applications and its relevant to the training cost. Please refer to the Weaknesses.

---

> ### Author Response · Authors · 2024-11-15
>
> First, thank you for recognizing the issues with the conventional evaluation protocol and for appreciating our efforts in identifying these issues and conducting extensive experiments.
>
> Our responses to the weaknesses you mentioned are outlined below.
>
> **1. Concerns on the proposed two-phase protocol**
>
> First, **we would like to emphasize that the high-level concept of this protocol (in Figure 2 of the manuscript) can be broadly applied across various domains of CL**. For instance, to account for imbalanced classes per task, a CL scenario in both phases can be designed based on this factor. Also, incorporating these factors is likely to amplify the differences between the CL scenarios in the two phases, leading us to expect that each algorithm will exhibit even poorer generalizability of its CL capacity compared to the results in the manuscript. In conclusion, we believe that the high-level concept of our two-phase protocol allows for assessing the generalizability of each algorithm’s CL capacity across other domains.
>
> **2. The experiments only consider class-incremental learning**
>
> As mentioned in the first paragraph of the Introduction section, we would like to reiterate that a flawed conventional evaluation protocol is still widely used across most domains. Given this, **we chose class-incremental learning as the representative domain, as it is not only one of the most actively researched areas but also considered highly challenging than task- and domain-incremental learning[1,2]**. Also, we conducted extensive experiments on numerous scenarios using 15 of the most prominent class-incremental learning algorithms, revealing that newer algorithms tend to be overestimated in the conventional evaluation protocol. **This finding suggests that similar issues could likely arise in other continual learning domains that also rely on conventional, flawed evaluation protocols**.
>
>
> [1] Three scenarios for continual learning, NeurIPSW 2018
>
> [2] A Comprehensive Survey of Continual Learning: Theory, Method and Application, TPAMI 2024
>
> **3. Most of the algorithms have similar trends**
>
> **We would like to reiterate our claim: “Under the revised evaluation protocol, newer algorithms tend to exhibit lower generalizability in terms of CL capacity or encounter issues such as inefficiency or instability, despite achieving high performance in the flawed conventional evaluation protocol.” (refer to Lines 516–525)**. In this context, please note once more the performance trends of the latest algorithms—FOSTER, BEEF, and MEMO—in Figure 4(b), where their performance differs markedly between $D^{HT}$ and $D^{E}$. Specifically, while model expansion-based methods like FOSTER and MEMO perform well on $D^{HT}$, they underperform on $D^{E}$ compared to methods like WA, PODNet, and BiC, which are order algorithms and maintain consistent model sizes. Additionally, although DER achieves high performance on both datasets, it has significant efficiency issues, as shown in Figure 6(b) and Lines 380–382.
>
> Furthermore, Figure 8(b) experimentally demonstrates how performance rankings shift considerably from $D^{HT}$ to the two $D^{E}$ datasets. For instance, EASE, one of the latest representation-based methods, shows strong performance on $D^{HT}$, but its performance on $D^{E}$ is lower than that of relatively simpler prompt-based methods. Additionally, while Ranpac exhibits strong results across all datasets, our experiments reveal it suffers from serious instability in certain scenarios (see Figure 10 and graphs in Appendix C.2).
>
> Additionally, note that Figures 5 and 9 again illustrate this trend, where recent algorithms perform significantly worse than earlier ones in many scenarios.
>
> Based on the above results, **we respectfully disagree with the reviewer’s statement that 'the advanced methods exhibit consistent advantages'**. We are eager to discuss this weakness further and engage in a constructive conversation.
>
> **4. The training cost is independent of the hyperparameter issue.**
>
> We would like to highlight the relationship between the proposed evaluation protocol and training costs. For instance, when utilizing cloud services with GPU resources, the cost is typically determined by usage time. The training duration for each algorithm (assuming the same model size and dataset) is primarily influenced by the number of computations required, including hyperparameter configurations—especially the number of epochs. To accurately assess each algorithm’s training cost (i.e., GPU usage time), it is crucial to compare training times using the optimal hyperparameters, which yield the best performance for each algorithm. **By employing our protocol to identify these optimal hyperparameters found from the hyperparameter tuning phase, we ensure a fair and effective comparison of each algorithm's efficiency in terms of training cost.**
>
> We respectfully ask the reviewer to consider these responses and look forward to an active discussion.

---

> ### Author Response · Authors · 2024-11-22
>
> Thank you for your valuable comments. Below is how we incorporated your feedback into the revised manuscript:
>
> 1. Concerns regarding the proposed two-phase protocol:
>     * We added a discussion on this concern in Lines 246–249 to address this.
> 2. Experiments only consider class-incremental learning:
>     * Additional explanations regarding the focus on class-incremental learning have been added in Lines 92–94 and 253–256. Conclusions derived from these results were included in Lines 524–526. To emphasize the use of class-incremental learning as a critique of the conventional evaluation protocol, we updated the title of the paper to "Hyperparameters in Continual Learning: A Reality Check with Class-incremental Learning”
> 3. Training cost being independent of hyperparameter issues:
>     * The relevant discussion was added in Lines 378–383.
>
> We believe these changes have clarified and strengthened our manuscript. Thank you again for your constructive feedback, and we look forward to further engaging discussions during the remaining review period.

---

> > ### Comment · Reviewer_irJH · 2024-11-25
> >
> > I thank the authors for their rebuttal. I appreciate the idea about checking the selection of hyperparameters in continual learning, it is an important problem for real applications. However, I believe this work has many aspects to improve, especially the generality of its setups and more comprehensive experiments. Therefore, I keep my rating unchanged.

---

> > > ### Author Response · Authors · 2024-11-27
> > >
> > > ### Gentle Reminder for Reviewer irJH
> > >
> > > Thank you once again for your valuable feedback on our work. We would like to kindly remind you that we have provided a response to your remaining concerns, along with a revised version of the manuscript, as noted above.
> > >
> > > With the revision deadline approaching, we would greatly appreciate it if you could take a moment to review our response and share any additional feedback or concerns at your earliest convenience. Your thoughtful comments have been instrumental in improving our work, and we want to ensure that we address any further suggestions you may have before the deadline. We look forward to continuing the discussion with you.
> > >
> > > Thank you.

---

> > > ### Author Response · Authors · 2024-12-01
> > >
> > > Dear Reviewer irJH,
> > >
> > > As the discussion period approaches its conclusion, we would like to kindly provide a gentle reminder. In our response, we have summarized our replies into two main points to address your remaining concerns. We sincerely hope this clarifies any misunderstandings and helps alleviate your concerns.
> > >
> > > We greatly value your critical comments and look forward to understanding the specific reasoning behind them, as well as receiving your detailed feedback on our responses. We are eager to engage in further constructive discussions based on your valuable insights.
> > >
> > > Thank you for your time and thoughtful review.

---

> ### Author Response · Authors · 2024-11-25
> **Response to Reviewer irJH**
>
> We sincerely thank the reviewer for their thoughtful comments and for engaging deeply with our work. We also greatly appreciate your acknowledgment of the importance of addressing the issues surrounding conventional evaluation protocols and hyperparameter selection methods in continual learning (CL). Your feedback has been invaluable in improving our manuscript.
>
> ----
>
> Despite our earlier response, we understand that there are remaining concerns regarding:
> 1. **The generality of the proposed evaluation protocol to other CL domains**, and
> 2. **The need for more comprehensive experiments across diverse CL domains**.
>
> We respectfully provide additional clarifications on these points for your consideration.
>
> ### 1. Generality of the Proposed Evaluation Protocol
> The primary contribution of our proposed Generalizable Two-phase Evaluation Protocol (GTEP) lies in its separation of hyperparameter tuning (Phase 1) and evaluation (Phase 2). This core concept, as detailed in Figure 1 and Lines 152–161 of the manuscript, aims to address the limitations of the conventional evaluation protocol, which evaluates algorithms on the same "seen" scenarios used for hyperparameter tuning. As demonstrated in Figure 3 and Lines 200–211, GTEP ensures that hyperparameters optimized in one phase are evaluated in a separate phase, leading to a more robust and realistic evaluation framework.
>
> **We would like to argue that the scenarios in both phases can incorporate complexities such as imbalanced classes per task, class imbalance within tasks, blurred task boundaries, or different task types, as suggested by the reviewer ("Generate a CL Scenario" in Figure 3). These variations are seamlessly adaptable within the high-level structure of GTEP.** Moreover, as noted in Lines 215–216, even under the assumption of shared CL scenario configurations across the two phases (albeit with different datasets), our experiments demonstrate a significant lack of generalization in the performance of state-of-the-art algorithms. Therefore, **incorporating these complexities into the two phases would likely amplify the observed shortcomings, further highlighting the limitations of the conventional protocol**.
>
> Given these points, we kindly request further clarification from the reviewer regarding any specific aspects of the proposed GTEP that may hinder its generality to other CL domains. Such insights would be invaluable in helping us address your concerns.
>
> ### 2. Comprehensive Experiments Across Diverse CL Domains
> Our study selected class-incremental learning (class-IL) as the focal domain due to its prominence in CL research. We evaluated 15 representative algorithms across diverse scenarios, conducting over 8,000 experiments. These results demonstrate that the conventional evaluation protocol systematically overestimates the CL capacity of these algorithms.
>
> We believe this extensive analysis substantiates the generalizability of our conclusions to other CL domains for the following reasons:
> 1. **Widespread adoption of the conventional protocol across CL domains**: As described in Lines 34–47, the conventional protocol is the de facto standard in various CL domains. Its deficiencies in hyperparameter tuning and evaluation methodology are not domain-specific.
> 2. **Task-specific differences do not alter core evaluation principles**: While individual tasks and their associated algorithms (e.g., semantic segmentation or self-supervised learning) may differ, the methodology for hyperparameter tuning and evaluation consistently adheres to the flawed protocol depicted in Figure 1. Consequently, although the extent of the performance gap between the conventional protocol and GTEP may vary across CL domains, the underlying issues persist.
>
> In this regard, our primary contribution lies in introducing GTEP as a realistic evaluation framework and demonstrating its necessity through extensive experimentation in class-IL. The previously obscured performance gaps revealed by our study, which we believe would generalize to other domains for the reasons outlined above, underscore the urgent need for a paradigm shift in CL evaluation. **Consequently, while additional experiments in other CL domains would be a valuable extension, we believe they are not essential to support the central claims of our paper. Therefore, we kindly ask the reviewer to consider whether the lack of such experiments critically undermines our main contributions**.
>
> We would greatly appreciate it if the reviewer could elaborate on why comprehensive experiments in other CL domains are deemed necessary. Such insights would be invaluable in helping us address your concerns more effectively.
>
> ----
>
> Once again, we sincerely thank the reviewer for their insightful feedback and the time invested in assessing our paper. We hope this additional clarification addresses your concerns and look forward to your thoughts.

---

### Official Review · Reviewer_PGuw · 2024-10-31

**Soundness:** 2
**Presentation:** 2
**Contribution:** 2
**Rating:** 3
**Confidence:** 5

**Summary:**

This paper aims to tackle the class-incremental learning problem, which is important to the machine learning field. The authors come up with a new evaluation protocol to investigate CIL methods of generalization. The authors have done extensive experiments to investigate the performance of different methods.

**Strengths:**

1.	This paper aims to tackle the class-incremental learning problem, which is important to the machine learning field.
2.	The topic of hyper-parameter robustness is interesting and has not been investigated in the CIL field
3.	The authors have done extensive experiments to investigate the performance of different methods.

**Weaknesses:**

1.	Although the authors have done extensive experiments in their new CIL setting, my major concern lies in the rationality of it. In typical machine learning scenarios, the training and testing data are i.i.d. sampled from the same training set. In other words, we train a model, evaluate it on the validation set, and utilize the best model to test on the test set (which has the same data distribution as the validation set). However, the authors advocate using the different data distributions for validation and testing, which is against common sense in typical machine learning. After reading the introduction, I would expect the authors to separate the original testing set into two disjoint sets for the current evaluation.
2.	Although the title is about continual learning, I find the experiments only focus on the class-incremental learning scenario. I would expect more interesting results in other continual learning settings like task-incremental learning, domain-incremental learning, learning with pre-trained vision-language models, etc.
3.	How to holistically evaluate a CIL algorithm has been also explored in another ICLR paper, i.e., [1], which extensively discusses the capability of different continual learning algorithms. In this aspect, this paper seems to advocate a typical case of CLEVA-Compass, making the contribution limited.
4.	The topic of this paper seems to be too narrow on the generalization ability, which is different from the typical CIL setting. I would suggest the authors name the protocol with some new name to avoid ambiguity.
5.  Finally, I also noticed a critical fact that leads to wrong conclusions. As the authors figure out from the main paper, DER is the most robust class-incremental learning algorithm. However, as they are using the PyCIL package, the reproduced DER is also not the full version, which does not implement the masking and pruning process in DER. See https://github.com/G-U-N/PyCIL/blob/31f2372d374c3f9a6c86d82b3c3ea4e0a880db63/models/der.py#L1C104-L1C124 (PyCIL's implementation), https://github.com/Rhyssiyan/DER-ClassIL.pytorch (DER official repo),
and
https://arxiv.org/pdf/2103.16788 (Eq.8 to Eq. 10). The main reason, I assume, is that the masking and pruning functions are also not robust and cannot be reproducible. Hence, using such code for comparison obviously leads to unfair comparisons among different methods.

[1] CLEVA-Compass: A Continual Learning EValuation Assessment Compass to Promote Research Transparency and Comparability. ICLR 2022

**Questions:**

See Weaknesses

---

> ### Author Response · Authors · 2024-11-15
>
> **1. The rationality of the proposed evaluation protocol.**
>
> **We would like to strongly argue that the evaluation method described by the reviewer may be common sense in some machine learning settings, but it is not a universal rule that applies to all learning scenarios**. Effective evaluation in machine learning should prioritize realistic methods tailored to each learning scenario, rather than rigidly adhering to assumptions (e.g., i.i.d.) for theoretical convenience. As one example in machine translation (MT), evaluation often focuses on the model's ability to generalize to unseen data, and one common approach is indeed to separate training and validation/test data based on their time of creation, which results in distinct distributions. This time-based split helps measure how well models perform on more recent language usage that might not appear in the training data, reflecting a more realistic scenario for deployment (please check more details in [1]).
>
> As outlined in Section 3.1, **we argue that the proposed evaluation protocol that separates the hyperparameter tuning and evaluation phases across different datasets offers a more realistic reflection of real-world continual learning scenarios (note that both Reviewer 3Qrs and irJH also support this protocol as a strength of our paper)**. We are eager to discuss this Weakness further and to engage in a constructive discussion.
>
> [1] ACL 2016 FIRST CONFERENCE ON MACHINE TRANSLATION (WMT16)
>
> **2. Considering other continual learning settings.**
>
> We would like to emphasize that it is challenging to cover all CL domains; therefore, **we focused our experiments on class-incremental learning, the most actively studied area in CL research[1,2]**. While we chose a single domain, we conducted an extensive evaluation on 15 of the most representative algorithms, covering the progression from earlier to the most recent methods in the field. Our experiments reveal that in the conventional evaluation protocol, newer algorithms tend to have their CL capacity overestimated across various scenarios. As stated in the first paragraph of the Introduction, **the conventional evaluation protocol is widely adopted across most CL domains, making it reasonable to infer that similar issues likely exist in other domains as well**.
>
> [1] Three scenarios for continual learning, NeurIPSW 2018
>
> [2] A Comprehensive Survey of Continual Learning: Theory, Method and Application
>
>
> **3. Comparing with another paper.**
>
> Thank you for introducing an interesting paper as relevant prior work. The primary contribution of this work is the proposal of CLEVA-Compass (Continual Learning EValuation Assessment Compass), a visual framework that enhances the evaluation and transparency of various methods in Continual Learning. However, we would like to point out that, despite the publication of that work two years ago, the conventional evaluation protocol remains dominantly used (see the first paragraph of our Introduction for further context). Also, we wish to highlight some key differences between that work and our own. Specifically: 1) we focus on constructing a specific revised evaluation protocol for accurate assessment, and 2) through extensive experimentation, we bring to light critical issues with the conventional evaluation protocol. Additionally, **we believe there is a significant distinction between discussing proper evaluation methods and presenting extensive experimental results that highlight the flaws in the conventional evaluation protocol**. In this regard, our paper falls into the latter category, offering a distinctly different contribution compared to the paper mentioned by the reviewer. We will cite this paper and incorporate the above discussion into the revised version of our manuscript.
>
> **4. New name of the proposed protocol.**
>
> We will assign a name to the proposed protocol to reduce ambiguity in a future update.
>
> **5. Implementation issues of DER**
>
> Thank you for your comments regarding the implementation of DER. As the reviewer mentioned, Neither PyCIL nor the official code of DER includes the implementation details for masking and pruning. However, **we would like to emphasize that, in the manuscript, we did not solely praise DER for achieving excellent performance. As highlighted in Figure 6(b) and (c) and in Lines 381-382, we pointed out the inefficiency from a parameter perspective due to the lack of pruning implementation**. We were aware of this implementation issue, but unfortunately, we overlooked adding this detail to the manuscript. We will be sure to include it in the future update. Additionally, **we would like to argue that our paper's main focus is not merely on showing that a particular algorithm achieves strong performance, but rather on exposing the issues in the conventional evaluation protocol through extensive experiments (see Sec. 5 of the paper)**.
>
> We hope that the reviewer will carefully consider these responses and engage in an active discussion.

---

> ### Author Response · Authors · 2024-11-22
>
> Thank you for your feedback. Below, we detail how your comments were incorporated into the revised manuscript:
>
> 1. Rationality of the proposed evaluation protocol:
>     * We have added a discussion addressing this concern in Lines 200–203.
> 2. Considering other continual learning settings:
>     * Additional explanations regarding the focus on class-incremental learning have been added in Lines 92–94 and 253–256. We also included conclusions derived from these results in Lines 524–526. To highlight the use of class-incremental learning to critique the conventional evaluation protocol, we updated the title to “Hyperparameters in Continual Learning: A Reality Check with Class-incremental Learning.”
> 3. Comparison with another paper:
>     * The mentioned paper has been cited, and relevant discussion was added in Lines 144–146.
> 4. New name for the proposed protocol:
>     * Following your suggestion, we have set a new name, the Generalizable Two-phase Evaluation Protocol (GTEP). (See Line 197)
> 5. Implementation issues of DER:
>     * Details on the implementation of DER were included in Lines 280–282. Additionally, to clarify that our work critiques the conventional evaluation protocol rather than advocating for specific algorithms, we revised Lines 533–534.
>
> We have worked diligently to address your feedback and believe that incorporating your comments has strengthened the manuscript. We look forward to your thoughts and further discussions.

---

> > ### Comment · Reviewer_PGuw · 2024-11-23
> >
> > I appreciate the authors’ efforts in the rebuttal. However, my concerns about the contributions are far from being solved. If this paper aims to raise the concern that current class-incremental learning algorithms are unstable, everyone agrees since it is a long-standing problem in the machine learning field --- but I do not think such a contribution is enough for ICLR. To me, the authors have done extensive experiments, while the results are poorly concluded. I would expect the comparison results when tuning the algorithms with the same validation set (that shares the same data distribution as the testing set). Besides, I would expect the authors to reproduce the results of DER with the masking and pruning stages since the paper implicitly indicates its robustness in the experiments. Considering these concerns are far from being addressed, I will maintain my rating.

---

> ### Author Response · Authors · 2024-11-23
> **Thank you for your response!**
>
> Thank you for your response and for sharing your evaluation criteria. After reviewing your comments, we believe there are still some misunderstandings regarding our work and that the basis for your negative evaluation remains unclear. We have prepared the following additional response to address these issues.
>
> ---
>
> **1. Objective and Contribution of This Paper**
>
> The primary focus of our paper is not to argue that class-incremental learning algorithms are unstable, but rather to highlight the limitations of the *conventional evaluation protocol* that is predominantly used in various continual learning (CL) domains. To reveal these limitations, we conducted extensive experiments in the most widely studied domain, class-incremental learning. This is explicitly discussed in Responses 1 and 5 and manuscript.
>
> ---
>
> **2. Regarding the "Long-Standing Problem" and Current CL Research**
>
> Even if the limitations of the conventional evaluation protocol are acknowledged as a "long-standing problem" by many researchers, as stated in the first paragraph of the Introduction, the reality is that this protocol continues to be dominantly used in the field. In this context, the greater issue lies in the current research landscape, where more emphasis is placed on designing state-of-the-art algorithms that achieve better performance under flawed evaluation protocols.  Also, **there is a significant distinction between merely recognizing a problem and proposing a revised evaluation protocol with extensive experiments**. Furthermore, **we believe that major conferences like ICLR provide space not only for papers introducing novel algorithms but also for contributions like ours, which point out and address fundamental evaluation challenges in the field**.
>
> ---
>
> **3. *"Results are poorly concluded"* – Evaluation protocol suggested by the reviewer**
>
> Regarding your comment:
> > *"I would expect the comparison results when tuning the algorithms with the same validation set (that shares the same data distribution as the testing set)."*
>
> We seek further clarification, as we interpret your statement in two potential ways:
>
> **3.1. Scenario 1:** Within a single CL scenario (as in Figure 1 of our paper), training, validation, and test datasets are sampled from the same distribution for each task.
> - In this case, even if validation and test datasets are separated, such an evaluation is unrealistic for real-world applications of CL. This would imply that the optimal hyperparameters for each algorithm are determined using the training/validation data from a specific CL scenario, and then tested on the test data of the same scenario. This setup assumes prior knowledge of the exact CL scenario to be encountered during the test phase, which is impractical.
>
> **3.2. Scenario 2:** In CL scenarios of both phases (as in Figure 4), training, validation, and test datasets were sampled from the same distribution for each task.
> - In this case, it is unnecessary to separate validation and test data for hyperparameter tuning and evaluation phases. This is because the unit of evaluation is the *CL scenario*, not individual datasets. In other words, the optimal hyperparameters are determined using a CL scenario (the "validation scenario") and applied to another CL scenario (the "test scenario") for evaluation. This approach is much more practical for real-world applications (as supported by Reviewer 3Qrs).
>
> Based on these considerations, we have provided detailed responses for both cases and explained our reasoning. **We kindly request further clarification regarding your comment and the basis for your expectation of comparisons under the mentioned evaluation setup**.
>
> ---
>
> **4. "Results are poorly concluded" – Results for DER**
>
> We are deeply concerned that the implementation and result of DER were cited as a major reason for rejecting our paper. Specifically, the claim that we reported superior performance for DER without properly implementing it—even when the official code itself is incomplete—is troubling.  As stated in our response, the goal of our paper is not to advocate for the superiority of DER or any specific algorithm. On the contrary, we explicitly highlight the issues with DER, such as its model size scaling linearly with the number of tasks. Moreover, our experiments focus primarily on uncovering the generalizability challenges faced by other algorithms, including FOSTER, Memo, and BEEF, which have been regarded as state-of-the-art under the conventional evaluation protocol. These support our central argument: the purpose of our paper is to critique the conventional evaluation protocol, not to endorse any particular algorithm.
>
> ---
>
> We hope this response resolves any misunderstandings and adequately addresses the concerns you raised. **We kindly request further discussion and encourage you to elaborate on the specific reasons or evidence behind your critical comments that led to the rejection of our paper.**
>
> Thank you.

---

> ### Author Response · Authors · 2024-11-27
>
> ### Gentle Reminder for Reviewer PGuw
>
> Thank you once again for your valuable feedback on our work. We would like to kindly remind you that we have provided a response to your remaining concerns, along with a revised version of the manuscript, as noted above.
>
> With the revision deadline approaching, we would greatly appreciate it if you could take a moment to review our response and share any additional feedback or concerns at your earliest convenience. Your thoughtful comments have been instrumental in improving our work, and we want to ensure that we address any further suggestions you may have before the deadline. We look forward to continuing the discussion with you.
>
> Thank you.

---

> > ### Author Response · Authors · 2024-12-01
> >
> > Dear Reviewer PGuw,
> >
> > As the discussion period approaches its conclusion, we would like to kindly provide a gentle reminder. In our response, we have summarized our replies into four main points to address your remaining concerns. We sincerely hope this clarifies any misunderstandings and helps alleviate your concerns.
> >
> > We greatly value your critical comments and look forward to understanding the specific reasoning behind them, as well as receiving your detailed feedback on our responses. We are eager to engage in further constructive discussions based on your valuable insights.
> >
> > Thank you for your time and thoughtful review.

---

### Official Review · Reviewer_3Qrs · 2024-11-02

**Soundness:** 3
**Presentation:** 2
**Contribution:** 3
**Rating:** 6
**Confidence:** 4

**Summary:**

The paper proposes a more rigorous evaluation protocol for continual learning methods, emphasizing generalization to unseen scenarios. In contrast to the traditional approach, where hyperparameter tuning and performance measurement occur on the same sequential dataset, often without separation between test and validation sets, the authors propose separate hyperparameter tuning and evaluation phases. While the configuration of the continual learning scenario is identical for both stages, each uses a different dataset. The authors evaluate a number of class-incremental learning algorithms using this framework. Based on a range of experiments, they conclude that most modern class-incremental learning algorithms fail to achieve their reported performance under the new evaluation protocol.

**Strengths:**

The main strength of the paper is the extensive experimental evaluation conducted under a rigorous evaluation protocol, leading to an important insight—the superior performance of some recent class-incremental methods may be due to meta-overfitting to the particular evaluation set through hyperparameter optimization. Challenging the dominant, flawed approach to evaluating continual learning algorithms is a valuable contribution that will hopefully help steer the community towards a more disciplined approach and help identify methods that have a good chance of generalizing to real-world applications.

**Weaknesses:**

Poor presentation and structure are the main weaknesses of the paper. Figure 4 (b) is perhaps the most important result, yet it is not given a prominent place. Figure 3 and Figure 7 could easily be short tables. Figure 1 and 2 should be simplified and would work together as a side-by-side comparison. Limiting the analysis to the 10-task and 20-task scenario, respectively, would allow to simplify Figures 5 and 9 and make them easier to parse. BEEF should be dropped from the figures (and, arguably, the analysis) if the authors were not able to run it. The hyperparameter sets in B.1 and B.2 would be easier to read as tables.

Another weakness is the use of the number of parameters and training time, which are not reliable proxies for efficiency, as explained in Dehghani et al. 2021 (The Efficiency Misnomer). For an efficiency metric in continual learning, see Roth et al. 2023 (A Practitioner's Guide to Continual Multimodal Pretraining).

**Questions:**

For BEEF, have you tried a different implementation or different seeds?

In Figure 4 (b), why do almost all methods perform better on the unseen scenario?

What criterium did you use to select the methods for evaluation? Is it their availability in PyCIL and PILOT?

---

> ### Author Response · Authors · 2024-11-15
> **Author response 1**
>
> Thank you for recognizing the issues of the conventional evaluation protocol in continual learning (CL) research and for acknowledging the extensive experiments we conducted using a revised protocol to bring these issues to light. We sincerely hope that our work contributes to broader discussions on this topic and encourages future CL research to pursue meaningful achievements through more rigorous evaluation.
>
> Below is our response to the Weaknesses raised by your review:
>
> 1. **Poor presentation.**
>
> Our initial goal in drafting the paper was to convey the following points in sequence: (1) highlight the limitations of the conventional evaluation protocol and introduce the revised protocol, (2) provide details on experimental settings and algorithms in each scenario (e.g., number of hyperparameters), (3) demonstrate the issues through experimental results across various scenarios, and (4) conduct additional comparative analyses. To ensure clarity, we included figures strategically and used bar graphs in the experimental section to facilitate comparisons at a glance. However, based on your feedback, we could recognize the need for further improvements to make our key points even more clear. In future updates, we will revise the figures and tables to enhance clarity and emphasize our main findings more effectively in the paper.
>
> 2. **Metrics to evaluate training cost.**
>
> Thank you for your insightful comments. The paper you referenced ([1]) raises an important point: relying on a single cost indicator (e.g., FLOPs, number of parameters, or training time) to compare the efficiency of models with different architectures can lead to varying trends depending on the chosen indicator. We completely agree with this perspective. However, **since most CL research employs the same model (e.g., regularization-based methods) or starts from the same model (e.g., model expansion-based methods), we believe that the cost indicators used in this paper (i.e., number of parameters and training time) can serve as a somewhat reasonable basis for comparing the training costs of different algorithms.** We acknowledge, though, that these indicators are not fully ideal for assessing efficiency in CL research. In this regard, we also appreciate the reviewer’s suggestion to use the Memory-Adjusted-FLOPs (MAFs) metric from [2] as a potential alternative for future efficiency evaluations. Should the paper be officially published and the code made available, we will consider incorporating MAFs in future work.
>
>
> [1] Dehghani et al. 2021 (The Efficiency Misnomer)
>
> [2] Roth et al. 2023 (A Practitioner's Guide to Continual Multimodal Pretraining)

---

> ### Author Response · Authors · 2024-11-15
> **Author response 2**
>
> **Q1) For BEEF, have you tried a different implementation or different seeds?**
>
> First, we would like to clarify that, consistent with all other experiments, **we reported the averaged results of five seeds for the experiments using BEEF**. As shown in Figure 5 in the manuscript, BEEF produced valid results on CIFAR-100; however, for ImageNet-100, we observed NaN errors arising from a specific seed (i.e., a particular task order). The table below presents the experimental results of BEEF with randomly selected hyperparameters across five different seeds.
>
> Hyperparameters: ep_160_milestone_3_lr_0.05_lr_decay_0.1_batch_128_w_decay_0.005_scheduler_cosine_fusion_ep_120_energy_w_0.001_logits_align_1.7
>
> |Acc / AvgAcc| Seed 0 | Seed 1 | Seed 2 | Seed 3 | Seed 4
> | -------- | -------- | -------- | -------- |-------- |-------- |
> | BEEF     | NaN     | 52.52 / 64.43     | NaN     |NaN     |53.04 / 65.15     |
>
> Note that for each seed, the task order was applied consistently across experiments using different algorithms, and only BEEF displayed a similar trend—namely, the occurrence of NaN values in specific seeds—even under different hyperparameter settings. **These results again demonstrate that BEEF shows instability in learning with certain task orders**.
>
> **Q2) In Figure 4 (b), why do almost all methods perform better in the unseen scenario?**
>
> This discrepancy may arise from differences in the datasets used across scenarios. Although ImageNet-100, used as $D^{HT}$, and ImageNet-100-2, used as $D^{E}$, contain the same number of classes, they consist of entirely different class labels. **Due to these dataset differences, even with identical hyperparameters, the final performance may vary across scenarios**. Nevertheless, we believe that comparing the performance rankings of algorithms on each dataset enables a meaningful comparison of each algorithm’s CL capacity in both phases.
>
> **Q3) What criterium did you use to select the methods for evaluation? Is it their availability in PyCIL and PILOT?**
>
> Our rationale for selecting the algorithms used in our study is as follows. First, we chose class-incremental learning (CIL) as the primary category of continual learning (CL) for evaluation, as it is widely recognized as **a more challenging category** compared to task- and domain-incremental learning [1]. Additionally, CIL has recently become **the most actively researched area** in CL [1,2]. Within CIL, different algorithms have been introduced based on whether they utilize pretrained models, prompting us to include a range of algorithms from earlier to more recent ones in a unified framework. Consequently, as **PyCIL and PILOT have successfully reproduced many CL algorithms**, we applied our proposed revised evaluation protocol to these codebases to conduct our experiments.
>
> [1] Three scenarios for continual learning, NeurIPSW 2018
>
> [2] A Comprehensive Survey of Continual Learning: Theory, Method and Application, TPAMI 2024
>
> Lastly, we have made every effort to thoroughly address all Weaknesses and Questions raised by the reviewer. We will upload the revised paper, reflecting these comments, as soon as the revisions are complete and notify you. We look forward to any further comments and an active discussion.

---

> ### Author Response · Authors · 2024-11-22
>
> Thank you for your comments. Below, we outline how we addressed your feedback in the revised manuscript:
>
> 1. Highlighting the key results in Figure 4(b):
>     * We agree that Figures 4(b) and 7(b) represent some of the most critical results in our paper. To reflect this, we created a new Figure 2 that focuses on these key findings. This new figure has been introduced at the end of the Introduction to emphasize the central issues with the conventional evaluation protocol highlighted in this paper.
> 2. Limiting the analysis to specific scenarios:
>     * Detailed descriptions of the task scenarios have been added in Lines 268–271 and 400–403. Additionally, we have emphasized the importance of considering these scenarios in the experiments.
> 3. Instability of BEEF:
>     * As demonstrated in our response, we observed clear instability in BEEF under certain seeds (i.e., task orders). Thus, we retained BEEF's results in the manuscript. Further experimental results, including those presented in our response, will be added to the Supplementary in a future update.
> 4. Figures 3 and 7:
>     * These figures have been moved to the Supplementary section. We also plan to update them as tables in a subsequent revision.
> 5. Hyperparameter sets:
>     * We agree that presenting the hyperparameter sets in table format would be helpful. This will be included in a future update.
> 6. Figures 1 and 2:
>     * While Figures 1 and 2 were retained, we revised the caption for Figure 2 to clarify that the hyperparameter tuning phase closely mirrors the conventional evaluation protocol, making the distinction between the conventional protocol and ours more explicit.
> 7. Why do most methods perform better in the unseen scenario?
>     * We have added relevant discussion in Lines 316–318.
> 8. Criteria for selecting evaluation methods:
>     * Additional details on the selection criteria for class-incremental learning have been added in Lines 92–94 and 253–256.
>
> We hope these revisions will prompt further discussions. Once again, we appreciate your insightful comments.

---

> > ### Comment · Reviewer_3Qrs · 2024-11-22
> > **Response to the Authors**
> >
> > Thank you for the reply and making changes to the manuscript. I do think that the paper reads better now.
> >
> > I still think the instability of BEEF might be caused by a faulty implementation. I think it would be best to either diagnose and fix the issue or drop the method from the analysis.
> >
> > While I don't share the concerns raised by other reviewers about limiting the analysis to class-incremental setting, I do think that the paper could have much higher impact if it proposed an accessible benchmark together with the new evaluation protocol (including an efficiency metric). I will maintain my original score.

---

> > > ### Author Response · Authors · 2024-11-22
> > > **Thank you for your comments on our response!**
> > >
> > > Thank you for your thoughtful comments on our response. We are glad that your valuable feedback has helped improve the clarity and quality of our manuscript.
> > >
> > > Below, we provide our responses to address the points raised in your additional comments.
> > >
> > > ---
> > >
> > > 1. **Response to additional comments on BEEF**
> > >
> > > Following your concerns regarding BEEF, we revisited our experiments to investigate the reported issues. Specifically, we consistently encountered NaN values when training BEEF on ImageNet-scale datasets using ResNet-18. To determine whether this issue stemmed from errors on our side or inherent instability in the algorithm, we conducted further investigations as follows:
> > >
> > > 1.1. **Hyperparameter Verification for BEEF**
> > >    We reviewed the hyperparameter values used in our experiments against those reported in the original paper. As outlined in Section C.2 of the Appendix in the BEEF paper, the paper reported using a learning rate of 0.1 (with a StepLR scheduler) and a mini-batch size of 256 when training on ImageNet with ResNet-18. **These values fall within the range of hyperparameters we considered during our experiments**.
> > >
> > > 1.2. **GitHub Issues Review**
> > >    We revisited GitHub repositories to check for reports of similar issues. Although the official BEEF codebase does not have an active Issues tab, we found a related discussion in the PyCIL repository (**notably, BEEF and PyCIL share the same authors**). In [Issue #64](https://github.com/G-U-N/PyCIL/issues/64) on the PyCIL GitHub, another user reported encountering the same NaN problem. A different user (not the author) who suffered from the similar problem suggested using a lower learning rate than the one reported in the paper as a potential solution.
> > >
> > > Based on these findings, we emphasize that the NaN issues persisted even when using hyperparameters consistent with those reported in the original paper. This aligns with similar reports from other users, suggesting that the instability might be an inherent limitation of BEEF. Therefore, we believe it is important to include these negative results in our paper. Additionally, we are currently experimenting with lower learning rates, as recommended in the Issue discussion. If results become available before the discussion concludes, we will share them promptly.
> > >
> > > ---
> > >
> > > **2. Proposing an accessible benchmark together with the new evaluation protocol**
> > >
> > > We have already included the code for the proposed evaluation protocol in the supplementary materials. In preparing the camera-ready version, we will refine and release it as an accessible benchmark and protocol. Furthermore, if the official code for the mentioned efficiency metric in your review becomes available, we will reference it and incorporate it into our final evaluation protocol code.
> > >
> > > ---
> > >
> > > Once again, we sincerely thank you for your constructive comments and your positive evaluation of our research.

---

> > > ### Author Response · Authors · 2024-11-27
> > >
> > > ### Gentle Reminder for Reviewer 3Qrs
> > >
> > > Thank you once again for your valuable feedback on our work. We would like to kindly remind you that we have provided a response to your remaining concerns, as noted above.
> > >
> > > With the revision deadline approaching, we would greatly appreciate it if you could take a moment to review our response and share any additional feedback or concerns at your earliest convenience. Your thoughtful comments have been instrumental in improving our work, and we want to ensure that we address any further suggestions you may have before the deadline. We look forward to continuing the discussion with you.
> > >
> > > Thank you

---

> > > > ### Comment · Reviewer_3Qrs · 2024-11-27
> > > > **Reply to the authors**
> > > >
> > > > Thank you for providing additional information about the method. To be clear—I believe that the authors did observe the lack of stability for BEEF and while trying different seeds or hyperparameters is a good idea to gauge how frequent the issue is, it doesn't help the reader understand where the instability is coming from.
> > > >
> > > > > Based on these findings, we emphasize that the NaN issues persisted even when using hyperparameters consistent with those reported in the original paper. This aligns with similar reports from other users, suggesting that the instability might be an inherent limitation of BEEF.
> > > >
> > > > It still might be a problem with the implementation in the BEEF codebase. It is crucial to understand where exactly the NaNs are coming from and what is causing them. It could be an easy fix of some minor numerical instability. If the authors claim that the method is inherently unstable, they need to at least hint at what is causing the instability.

---

> ### Author Response · Authors · 2024-11-25
>
> Hello, Reviewer 3Qrs,
>
> To further validate the instability of BEEF, we conducted additional experiments with lower learning rates, as suggested in the issues section of its official GitHub repository. We set the learning rate to [0.001, 0.005, 0.01, 0.015, 0.02] while keeping the other hyperparameter sets consistent with the original settings described in the manuscript.
>
> Using the ImageNet-100 dataset, we tested 20 randomly sampled hyperparameter configurations, and unfortunately, we could not find a single configuration that avoided NaN values across all seeds. Below, we present the results for two representative hyperparameter settings which achieve relatively better performance in some seeds:
>
> * **Hyperparameters**
> HP1: `ep_160_milestone_2_lr_0.01_lr_decay_0.5_batch_256_w_decay_0.005_scheduler_cosine_fusion_ep_160_energy_w_0.01_logits_align_2.3`
> HP2: `ep_200_milestone_2_lr_0.01_lr_decay_0.1_batch_256_w_decay_0.005_scheduler_cosine_fusion_ep_160_energy_w_0.005_logits_align_1.1`
>
> | **Acc / AvgAcc** | **Seed 0** | **Seed 1** | **Seed 2** | **Seed 3** | **Seed 4** |
> |------------------|------------|------------|------------|------------|------------|
> | **BEEF (HP1)**   | 49.52 / 65.57 | 49.24 / 59.22 | NaN | NaN | NaN |
> | **BEEF (HP2)**   | 48.62 / 65.14 | 46.40 / 58.31 | NaN | NaN | NaN |
>
> In our initial response, we mentioned that NaN was observed for all seeds except Seed 1 and Seed 4. However, when the learning rate was lowered, all seeds (except Seed 0 and Seed 1) produced NaN results. Despite using the reported hyperparameters from the paper and the learning rates mentioned in the GitHub Issue (remember that both BEEF and PyCIL share the same authors), we observed instability on certain seeds (i.e., specific task orders), reaffirming that this instability is inherent to the algorithm rather than an implementation issue. We believe this instability issue is worth reporting in the paper.
>
> We plan to include these findings in the Appendix in a future update. Once again, we appreciate your thoughtful review and valuable comments on our paper.
>
> Thank you!

---

> ### Author Response · Authors · 2024-11-28
>
> Thank you once again for your detailed and valuable comments. We share your perspective, and to address your concerns, we have been conducting an in-depth analysis based on the PyCIL implementation of BEEF to identify the root cause of its instability. Our goal is to include a dedicated section in a future update that sheds light on the source of this issue. Through various ablation studies and experiments, **we could confirm that the instability in BEEF is not due to minor numerical issues that can be easily resolved**. Below, we summarize our findings:
>
> 1. **Dataset Scale Dependency**: We observed no issues when training on the CIFAR dataset; however, NaNs consistently occur when training at the ImageNet scale under certain seeds (specific task orders).
> 2. **Seed-Dependent Variability**: Our experiments confirmed that the NaN occurrences are not tied to specific tasks or iterations. Instead, the occurrence timing varies across different seeds.
> 3. **Adversarial Learning Process**: The root cause is not related to numerical instability in loss functions but rather stems from the adversarial learning process used to generate samples, which is a core component of the BEEF algorithm.
>
> As seen in the PyCIL BEEF implementation ([Line 408 of BEEF Code in PyCIL](https://github.com/G-U-N/PyCIL/blob/0cb8ad6ca6da93deff5e8767cfb143ed2aa05809/models/beef_iso.py#L408C9-L408C17)), BEEF employs adversarial learning during training each task to generate samples, which are subsequently used to calculate the proposed energy loss. Our analysis revealed that the adversarial examples (e.g., `embedding_k` in the code) increasingly amplify the feature map values of the copied model (`_network_copy` in the code) when passed through it over iterations. This leads to extreme value growth in the feature maps in certain cases, which subsequently causes NaN issues in the training model when these adversarial examples are used to compute the energy loss.
>
> We suspect that this issue arises from generating adversarial examples without applying any constraints. To mitigate this, we could explore introducing constraints such as L2 or L1 regularization to limit the extent of transformations during adversarial example generation. **However, we emphasize that this is not merely a numerical instability issue; it represents an algorithmic instability**. Additionally, it remains uncertain whether implementing such constraints would preserve BEEF’s reported performance on datasets like CIFAR and ImageNet under normal conditions.
>
> We sincerely appreciate your insightful comments, which have been instrumental in deepening our understanding of this issue. To report this issue, we will summarize the above findings in an appendix subsection in a future update to provide a comprehensive analysis of BEEF’s instability. Thank you for facilitating this valuable discussion, and please do not hesitate to share any further feedback during the remaining discussion period.
>
> Thank you.

---

> > ### Comment · Reviewer_3Qrs · 2024-11-28
> > **Response to the Authors**
> >
> > Thank you for coming back to me with a more detailed analysis that does point more towards a deeper problem with the algorithm. I think introducing constraints to limit the magnitude of transformations is a good idea. I am also a bit suspicious about the plain copy in line 412 (as opposed to deepcopy).
> >
> > Thank you again for engaging in discussion.

---

> > > ### Author Response · Authors · 2024-11-28
> > >
> > > Thank you for your positive feedback and for taking the time to provide a response. We sincerely appreciate your active participation in the discussion. Incorporating your comments has significantly improved the clarity and overall quality of our paper. For the final camera-ready version, we plan to further refine our paper comprehensively.
> > >
> > > We are truly grateful for your efforts.

---

### Author Response · Authors · 2024-11-15
**Complete to upload first responses**

We would like to express our sincere gratitude to all the reviewers for their valuable feedback. We have provided detailed responses to all the weaknesses and questions raised by the reviewers, making a particular effort to clarify any misunderstandings about the content of the paper. We kindly ask the reviewers to review our responses and look forward to receiving their feedback. We hope to engage in an active discussion.

Additionally, we are currently in the process of updating the paper to reflect the reviewers' comments. Once the updates are complete, we will upload the revised version and share another public comment to notify everyone.

Thank you.

---

### Author Response · Authors · 2024-11-22
**Revision Uploaded**

We would like to once again express our sincere gratitude to all the reviewers for their thoughtful comments and valuable feedback. We have completed and uploaded a revised version of our manuscript, incorporating the comments raised by the reviewers. The revisions made in response to each reviewer’s comments have been organized and detailed separately for each reviewer. We hope that our responses and the revised manuscript will encourage further productive discussions.

Thank you.

---

### Note · Authors · 2025-01-22

**Comment:**

We would like to express our gratitude to all the reviewers and ACs for engaging in meaningful discussions. Based on the feedback we received, we will further improve our paper.

**Withdrawal Confirmation:**

I have read and agree with the venue's withdrawal policy on behalf of myself and my co-authors.